# Methane emissions are predominantly responsible for record-breaking atmospheric methane growth rates in 2020 and 2021

**Liang Feng[1], Paul I. Palmer[1,2], Robert J. Parker[3,4], Mark F. Lunt[2], and Hartmut Bösch[3,4]**

[1]National Centre for Earth Observation, University of Edinburgh, Edinburgh, UK
[2]School of GeoSciences, University of Edinburgh, Edinburgh, UK
[3]National Centre for Earth Observation, Space Park Leicester, University of Leicester, Leicester, UK
[4]Earth Observation Science, School of Physics and Astronomy, University of Leicester, Leicester, UK

**Correspondence:** Paul I. Palmer (paul.palmer@ed.ac.uk)

**Abstract.** CE1 The global atmospheric methane growth rates reported by NOAA for 2020 and 2021 are the largest since systematic measurements began in 1983. To explore the underlying reasons for these anomalous growth rates, we use newly available methane data from the Japanese Greenhouse gases Observing SATellite (GOSAT) to estimate methane surface emissions. Relative to baseline values in 2019, we find that a significant global increase in methane emissions of $27.0 \pm 11.3$ and $20.8 \pm 11.4$ Tg is needed to reproduce observed atmospheric methane in 2020 and 2021, respectively, assuming fixed climatological values for OH. We see the largest annual increases in methane emissions during 2020 over Eastern CE2 Africa ($14 \pm 3$ Tg), tropical Asia ($3 \pm 4$ Tg), tropical South America ($5 \pm 4$ Tg), and temperate Eurasia ($3 \pm 3$ Tg), and the largest reductions are observed over China ($-6 \pm 3$ Tg) and India ($-2 \pm 3$ Tg). We find comparable emission changes in 2021, relative to 2019, except for tropical and temperate South America where emissions increased by $9 \pm 4$ and $4 \pm 3$ Tg, respectively, and for temperate North America where emissions increased by $5 \pm 2$ Tg. The elevated contributions we saw in 2020 over the western half of Africa ($-5 \pm 3$ Tg) are substantially reduced in 2021, compared to our 2019 baseline. We find statistically significant positive correlations between anomalies of tropical methane emissions and groundwater, consistent with recent studies that have highlighted a growing role for microbial sources over the tropics. Emission reductions over India and China are expected in 2020 due to the Covid-19 lockdown but continued in 2021, which we do not currently understand. To investigate the role of reduced OH concentrations during the Covid-19 lockdown in 2020 on the elevated atmospheric methane growth in 2020–2021, we extended our inversion state vector to include monthly scaling factors for OH concentrations over six latitude bands. During 2020, we find that tropospheric OH is reduced by $1.4 \pm 1.7$ % relative to the corresponding 2019 baseline value. The corresponding revised global growth of a posteriori methane emissions in 2020 decreased by 34 % to $17.9 \pm 13.2$ Tg, relative to the a posteriori value that we inferred using fixed climatological OH values, consistent with sensitivity tests using the OH climatology inversion using reduced values for OH. The counter statement is that 66 % of the global increase in atmospheric methane during 2020 was due to increased emissions, particularly from tropical regions. Regional flux differences between the joint methane–OH inversion and the OH climatology inversion in 2020 are typically much smaller than 10 %. We find that OH is reduced by a much smaller amount during 2021 than in 2020, representing about 10 % of the growth of atmospheric methane in that year. Therefore, we conclude that most of the observed increase in atmospheric methane during 2020 and 2021 is due to increased emissions, with a significant contribution from reduced levels of OH.

## 1 Introduction

The atmospheric growth rate of methane in the 21st century has defied a definitive explanation: following a period of near-zero growth during 2000–2007 (Rigby et al., 2008), growth rates have accelerated, with values reported by NOAA for 2020 ($15.19 \pm 0.41$ ppb) and 2021 ($18.12 \pm 0.47$ ppb) exceeding all prior values since their records began in 1983. The underlying reasons for these anomalous growth rates in 2020 and 2021 are currently subject to intense debate with some studies attributing most of the growth in 2020 to a reduction in the hydroxyl radical (OH) sink of methane due to global-scale reductions in nitrogen oxides due to pandemic-related industry shutdowns (Laughner et al., 2021). On the face of it, this appears to be a reasonable explanation, but recent studies have used satellite observations of atmospheric methane to reveal regional hotspots over the tropics that are responding to changes in climate and have global significance (Pandey et al., 2021; Lunt et al., 2019, 2021a; Pandey et al., 2017; Feng et al., 2022b; Palmer et al., 2021; Wilson et al., 2021). Here, we use satellite observations of methane from the Japanese Greenhouse gases Observing SATellite (GOSAT) to document global and regional changes in emissions, extending a recent study (Feng et al., 2022b). In the next section, we describe the data and methods used to infer methane emissions. In Sect. 3, we describe our results and conclude the study in Sect. 4.

## 2 Data and methods

We closely follow the methodology from a recent study (Feng et al., 2022b) in which we simultaneously infer methane and $CO_2$ fluxes in 2020 and 2021 by directly assimilating proxy GOSAT $XCH_4 : XCO_2$ retrievals (X denotes atmospheric column-averaged dry-air mole fraction). These data are anchored by surface methane and $CO_2$ measurements from an in situ observation network. The main advantage of this approach is that it does not rely on assumed model $CO_2$ concentrations to extract $XCH_4$ from the proxy ratio. For the sake of brevity, we only include details relevant to the calculations shown here.

### 2.1 GOSAT methane proxy data

We use version 9.0 of the proxy GOSAT $XCH_4 : XCO_2$ retrievals from the University of Leicester (Parker et al., 2020; Parker and Boesch, 2020), including both nadir observations over land and glint observations over the ocean. Analyses have shown that these retrievals have a bias of 0.2 %, with a single-sounding precision of $\sim 0.72$ %. We globally remove a slightly larger 0.3 % bias from the GOSAT proxy data to improve the comparison with independent in situ observations (Feng et al., 2017, 2022a). We assume that each single GOSAT proxy $XCH_4 : XCO_2$ ratio retrieval has an uncertainty of 1.2 % to account for possible model errors, including the errors in atmospheric chemistry and transport, which helps to prevent model overfitting to observations.

### 2.2 In situ data

To anchor the GOSAT proxy ratio observations (Fraser et al., 2014), we also simultaneously ingest the $CO_2$ and methane mole fraction data at surface-based sites, chosen from the NOAA compilation of the multi-laboratory in situ measurements (Di Sarra et al., 2021, 2022a; Cox et al., 2021, 2022b). We include the same subset of the surface sites used by a recent study that documented year to year variations of methane emissions during 2010–2019 (Feng et al., 2022b). We assume uncertainties of 0.5 ppm and 8 ppb for these in situ observations of $CO_2$ and methane, respectively (Feng et al., 2022b). We take advantage of the latest $CO_2$ (GLOB-ALVIEWplus v8.0 ObsPack) (Cox et al., 2021, 2022a) and methane (GLOBALVIEWplus v5.0 ObsPack) (Di Sarra et al., 2022a, 2021) data products to study 2020 and 2021.

### 2.3 GEOS-Chem atmospheric chemistry transport model

We use the GEOS-Chem model of atmospheric chemistry and transport at a horizontal resolution of $2°$ (latitude) $\times 2.5°$ (longitude), driven by the MERRA2 (Modern-Era Retrospective Analysis for Research and Applications, version 2) meteorological reanalyses from the Global Modeling and Assimilation Office (GMAO) Global Circulation Model based at NASA Goddard Space Flight Center.

Our $CO_2$ and methane model calculations are described in a recent study (Feng et al., 2022b). The a priori $CO_2$ flux inventory includes monthly biomass burning emission (Van der Werf et al., 2017); monthly fossil fuel emissions for 2019 in the absence of more recent data (Oda and Maksyutov, 2021); monthly climatological ocean fluxes (Takahashi et al., 2009); and 3-hourly terrestrial biosphere fluxes (Randerson et al., 1996).

The a priori methane fluxes from nature include monthly wetland emissions, including rice paddies (Bloom et al., 2017); monthly fire methane emissions (Van der Werf et al., 2017); and termite emissions (Fung et al., 1991). We include emissions from geological macroseeps (Kvenvolden and Rogers, 2005; Etiope, 2015). For a priori anthropogenic emissions, we use the EDGAR v4.32 global emission inventory for 2012 (Janssens-Maenhout et al., 2019) that includes various sources related to human activities (e.g. oil and gas industry, coal mining, livestock, and waste).

We use monthly 3-D fields of OH, consistent with observed values for the lifetime of methyl chloroform, from the GEOS-Chem full chemistry simulation (Mao et al., 2013; Turner et al., 2015) to describe the main oxidation sink of methane. Using pre-computed fields of OH greatly simplifies our calculations. We examine the sensitivity of our re-

sults to different OH distributions, as described below. Additionally, in the next section, we describe a joint methane–OH inversion experiment from which we also report results. We also include the net microbial consumption of methane in soil (Fung et al., 1991) and reaction with chlorine atoms (Thanwerdas et al., 2019).

To explore the sensitivity of our methane emission estimates for 2020 resulting from inferred reductions in OH due to large-scale industrial shutdown due to Covid-19 (Cooper et al., 2022), we also report a posteriori methane emission estimates that assume two different OH distributions, guided by observed changes in combustion and in tropospheric ozone. These sensitivity tests should not be considered as rigorous as our joint methane–OH inversion described below, since they represent a useful sanity check for our understanding.

First, we scale down our baseline monthly 3-D OH fields by 5 %, where combustion emissions of $CO_2$ (Oda and Maksyutov, 2021) were larger than the mean emissions over Africa, resulting in reductions mainly between 15 and 65° N. Our choice of 5 % represents a reduction based on a recent study (Laughner et al., 2021). A recent study that accounted for reductions in nitrogen oxide emissions estimated a global OH reduction of $\sim 4\%$ due to the Covid-19 lockdown in 2020 (Miyazaki et al., 2021), which showed strong spatial and temporal variations, with localized reductions peaking at 20 %–30 %. In the absence of direct measurements of OH and without considering co-reductions in non-methane hydrocarbons, these (and similar) results have large uncertainties.

Second, we assume a temporal–spatial distribution to describe the OH reduction in 2020, following a recent study on tropospheric ozone changes in 2020 and 2021 (Ziemke at al., 2022). First, we divide the world into regions: Northern Hemisphere (20–90° N) and the rest of the world. We assume that the reduction in OH in the Northern Hemisphere starts from the boreal spring of 2020 and peaks during the summer with a magnitude of 9 % (blue line, Fig. A1 in Appendix A), higher than the ozone reduction found by Ziemke et al. (2022). For latitudes south of 20° N, the time evolution of the reduction in ozone is less clear (Ziemke et al., 2022). For simplicity, we assume that the OH reduction at these latitudes (red line, Fig. A1) has a smaller peak value ($-2.3\%$) and with a time lag of 1 month compared to the region in the Northern Hemisphere.

## 2.4 Ensemble Kalman filter inverse method

We use an ensemble Kalman filter (EnKF) framework to simultaneously estimate $CO_2$ and methane fluxes from satellite measurements of the atmospheric $CO_2$ and methane (Feng et al., 2022b). Our state vector includes monthly scaling factors for 487 regional pulse-like basis functions (Fig. A2) that describe $CO_2$ and methane fluxes, including 476 land regions and 11 oceanic regions. We define these land sub-regions by dividing the 11 TransCom–3 land regions into 42 nearly equal sub-regions, with the exception for temperate Eurasia that has been divided into 56 sub-regions due to its large landmass. We use the 11 oceanic regions defined by the TransCom–3 experiment. We use a 4-month moving lag window to reduce the computational costs for projecting the flux perturbation ensemble into observation space long after ($> 4$ months) their emissions, beyond which time it is difficult to distinguish between the emitted signal from variations in the ambient background atmosphere (Fraser et al., 2014; Feng et al., 2017). Our a priori fluxes are described above. For simplicity, we assume a fixed uncertainty of 40 % for coefficients corresponding to the a priori $CO_2$ fluxes over each sub-region, and a larger uncertainty (60 %) for the corresponding methane emissions. We also assume that a priori errors for the same gas are correlated with a spatial correlation length of 300 km and a temporal correlation of 1 month.

As a sensitivity test, we also report methane fluxes inferred using the same EnKF approach but using the proxy GOSAT $XCH_4$ data and in situ methane data. These GOSAT $XCH_4$ retrievals are calculated from the $XCH_4 : XCO_2$ ratio by applying an ensemble mean of model $XCO_2$ and then bias-corrected according to comparison with Total Carbon Column Observing Network (TCCON) $XCH_4$ retrievals (Parker et al., 2020).

Currently, there is no direct observation of the global distribution of atmospheric OH. Indirect constraints on atmospheric OH from the changing lifetimes of trace gases such as CO and methane are insufficient to determine 3-D distributions of OH. Here, we extend the inversion state vector to simultaneously infer methane emissions (as described above) and six OH scaling factors for a priori monthly 3-D OH fields from the atmospheric methane observations (Sect. 2). These scaling factors correspond to six latitude bands: 75–50° S, 50–25° S, 25–0° S, 0–25° N, 25–50° N, 50–75° N. We do not consider scaling polar OH values. This calculation complements the OH sensitivity experiments described in the previous section. The Jacobian matrix, which describes the sensitivity of model methane concentrations to regional OH fields, is calculated with GEOS-Chem forced by a priori methane fluxes from the control run (Table 1) but with the OH climatology reduced by 5 % for each of the six regions. To reduce the computational cost of this calculation, we use the same 4-month lag window for the OH scaling factor estimates as for the methane emission estimates. This is so that each monthly OH scaling factor will be constrained only by observations in the subsequent 4 months, but its impact will remain for the entire experimental period. We used the OH climatology as our a priori and assume a uniform 3 % uncertainty for each of the six regions so that the 2-sigma range covers possible OH changes that span $\pm 6\%$. We use such a simple linearization scheme to adjust surface methane emissions and the monthly tropospheric OH by optimally fitting model calculations to atmospheric methane observations. We conduct the joint inversion for 2018 to 2021, including the six monthly OH scal-

ing factors and methane emissions, and use the same atmospheric methane data used by the control calculation.

## 2.5 Correlative data

To help interpret the changes in methane emission estimates we use additional datasets that are relevant to microbial or pyrogenic production of methane. We use monthly surface temperature fields at a spatial resolution of $2° \times 2.5°$ from the Modern-Era Retrospective Analysis for Research and Applications, version 2 (MERRA2) developed by the Global Modeling and Assimilation Office at the NASA Goddard Space Flight Center (Bosilovich et al., 2015). Precipitation data are taken from the NOAA CMAP (CPC Merged Analysis of Precipitation) long-term global rainfall dataset (Xie and Arkin, 1997) that provides near-global monthly coverage at a spatial resolution of $2.5° \times 2.5°$, from 1979 to near present. In addition, we use monthly total water storage (liquid water equivalent depth, LWE) on a $1° \times 1°$ global grid from the NASA–DLR Gravity Recovery and Climate Experiment Follow-on (GRACE-FO) (Landerer et al., 2020). Finally, we explore monthly biomass burning emissions from the Global Fire Emissions Database (GFED v4) (Van der Werf et al., 2017).

## 3 Results

Table 1 summarizes our global emission estimates inferred from GOSAT for 2020, 2021, and 2019, which we use as our baseline year throughout this study (Fig. A3). The largest change in our global a posteriori emissions, corresponding to the OH climatology, occurs during 2019–2020 (27.0 Tg) from 583.7 to 610.7 Tg yr$^{-1}$. Our a posteriori emission estimates for 2019 and 2020 are within 2 % of values reported by an independent study (Qu et al., 2022), consistent with our reported uncertainties. These elevated emissions are sustained, but not further increased, during 2021 (604.5 Tg yr$^{-1}$).

The 27.0 Tg emission increase in 2020 and the lack of further emissions growth in 2021 may appear inconsistent with the NOAA global annual mean growth rates of $15.19 \pm 0.41$ and $18.12 \pm 0.47$ ppb in 2020 and 2021, respectively (Table 1). Based on these reported atmospheric growth rates, and after considering the effects of methane sinks, we find that a one-box model calculation predicts an increase in emissions of 12.6 Tg between 2019 and 2020 and a further 15.1 Tg increase in 2021 (see Appendix B). These calculations use annual mean values that effectively represent emissions increase between the middle of each successive year rather than the beginning and end. After considering the increases in monthly mean NOAA data, we find that the simple box model predicts a similar increase in emissions between December 2019 (583.7 Tg yr$^{-1}$) and December 2020 (610.1 Tg yr$^{-1}$) of 26.4 Tg yr$^{-1}$, with emissions stabilizing thereafter, with mean emissions of 610.1 Tg yr$^{-1}$ in 2021.

The resulting a posteriori model atmospheric methane concentrations agree well with the assimilated in situ data, as expected, but also reproduce the spatial and temporal variations of methane reported by the independent TCCON measurement network. As such, we conclude that the global mean emission results inferred from GOSAT are consistent with those inferred from NOAA surface data over multi-year periods, assuming a fixed methane atmospheric lifetime.

Figure 1b shows the broad geographical breakdown for our reported global changes in methane emissions. Relative to 2019, we find widespread increased emissions during 2020, except for China and India. Relative to baseline values in 2019, we see the largest annual increases in methane emissions during 2020 over Eastern Africa ($14 \pm 3$ Tg), tropical South America ($5 \pm 3$ Tg), tropical Asia ($3 \pm 3$ Tg), and temperate Eurasia ($3 \pm 3$ Tg), and the largest reductions are observed over China ($-6 \pm 3$ Tg) and India ($-2 \pm 3$ Tg). We find comparable emission changes in 2021, relative to 2019, except for tropical and temperate South America where emissions increased by $9 \pm 4$ and $4 \pm 3$ Tg, respectively, and for temperate North America where emissions increased by $5 \pm 2$ Tg. Our results are broadly consistent with a recent study which showed that methane emissions inferred from TROPOMI were significantly higher in the first half of 2020 than during 2019 (McNorton et al., 2022). This study focused mainly on major countries, while we find the largest changes are over tropical latitudes where emissions in the second half of 2020 make significant contributions (Fig. A4).

Figure 2 shows the distribution of methane emissions from 2020 and 2021 and the relative changes from our 2019 baseline year (Fig. A3a). During 2020, there are significant decreases (20 %–30 %) over the manufacturing regions such as eastern China, India, central America, and eastern Europe. There are also significant increases across Eastern Africa (30 %–40 %), eastern North America (30 %), and maritime Southeast Asia (30 %). During 2021, we see similar changes in emissions, but they are typically exaggerated. There is more of a pronounced increase over Eastern Africa ($> 50$ %), southern Brazil (50 %), and eastern North America (up to 40 %), and large decreases are observed over eastern China ($-50$ %) and western Russia ($-50$ %). During 2021, there is also a large decrease over equatorial West Africa and eastern Europe (Fig. 1b). There are substantial seasonal changes in methane emissions (Fig. A4) that are broadly consistent with seasonal changes in temperature and rainfall (not shown). Using methane columns determined by the proxy data, assuming model values for $CO_2$ (Parker et al., 2020), we find results for 2020 and 2021 that are generally within 10 % of the values we report using the proxy data directly (Figs. A5 and A6).

Figure 3 shows different annual surface temperature warming patterns in 2020 and 2021. During 2020, the high northern latitudes are dominated by summer warming over Siberia (2–3 K on an annual scale) that has been linked to greenhouse gas emissions (Ciavarella et al., 2021), and sur-

**Table 1.** Global annual emission estimates of methane (Tg yr$^{-1}$) inferred from GOSAT (2019–2021) and in situ (2019–2020) atmospheric measurements of methane. The annual atmospheric methane growth rate (ppb yr$^{-1}$) for 2019 to 2022 reported by NOAA is also shown.

| | Global annual methane emissions (Tg yr$^{-1}$) | | |
| --- | --- | --- | --- |
| | 2019 | 2020 | 2021 |
| GOSAT methane inversion | $583.7 \pm 11.2$ | $610.7 \pm 11.3$ | $604.5 \pm 11.4$ |
| GOSAT methane–OH inversion | $585.3 \pm 13.1$ | $603.2 \pm 13.2$ | $603.7 \pm 13.2$ |
| Corresponding OH change (%) | $+0.91 \pm 1.7\%$ | $-0.52 \pm 1.7\%$ | $+0.62 \pm 1.7\%$ |
| In situ | $588.9 \pm 18.1$ | $601.4 \pm 18.6$ | – |
| NOAA atmospheric growth rate (ppb yr$^{-1}$) | $9.67 \pm 0.60$ | $15.19 \pm 0.41$ | $18.12 \pm 0.47$ |

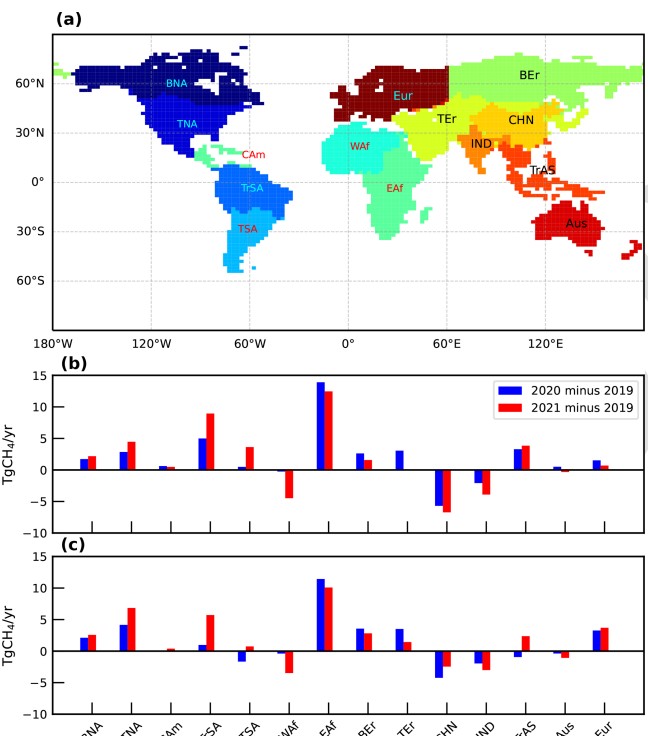

**Figure 1. (a)** Large-scale geographical regions for which we report methane changes (TgCH$_4$ yr$^{-1}$) in 2020 and 2021. **(b)** Differences between a posteriori emissions from 2020 and 2021 relative to inversion-specific baselines for 2019. Geographical regions, informed by TransCom–3 experiments (Gurney et al., 2004), include boreal North America (BNA), temperate North America (TNA), central America (Cam), tropical South America (TrSA), temperature South America (TSA), Europe (Eur), Western Africa (WAf), Eastern Africa (EAf), boreal Eurasia (BEr), temperate Eurasia (TEr), India (IND), China (CHN), tropical Asia (TrAs), and Australia (Aus). Panel **(c)** is the same as **(b)** but for a posteriori methane emission from the joint methane–OH inversion.

face temperatures over Alaska were 2–3 K cooler than baseline values in 2019, where there were comparatively small changes in groundwater ($< 5$ cm). North America, western Europe, and Scandinavia also experienced anomalously warm annual mean temperatures (typically within $\pm 2$ K of 2019 values). There were smaller changes in temperatures at low latitudes (typically $\pm 1$ K of 2019 values), but larger increases in groundwater ($\pm 10$–$20$ cm) associated with higher changes in rainfall (Fig. A7), particularly over Eastern Africa and eastern Brazil. During 2021, high northern latitudes were cooler than 2019 ($< 2$–$3$ K), except for the contiguous US and Canada (higher than 2019 values by 2–3 K). Midlatitudes and low latitudes generally did not experience the warm temperatures of 2020. Elevated groundwater was sustained in 2021 over Eastern and southern Africa, eastern tropical South America (principally Brazil but stretching up to Venezuela), central America, India, maritime Southeast Asia, and North and Southeast Australia. Groundwater decreased over the contiguous US, part of tropical South America, and parts of Eurasia. We find generally stronger annual and seasonal relationships between methane emission anomalies and hydrological anomalies (rainfall and groundwater) for 2020 and 2021 (Fig. 3) than for temperature anomalies. Particularly, we find statistically significant large-scale positive correlations (typically 0.6–0.9; $p < 0.001$) for all seasons between methane and groundwater anomalies over Eastern Africa, tropical South America, and tropical Asia (Fig. 4), but there is no significant correlation between methane and surface temperature anomalies (not shown). This is consistent with recent studies that have highlighted an increasing role for microbial sources in the tropical methane budget (Lunt et al., 2019; Feng et al., 2022b; Wilson et al., 2021). Over North America, we find a significant negative correlation (from $-0.3$ to $-0.6$; $p < 0.001$) with rainfall during MAM and JJA and a significant positive correlation with temperature during JJA (0.4; $p < 0.001$), which we do not currently understand. Fire emissions did not increase much where we report the largest increases in methane emissions in 2020 or 2021, except over central Canadian provinces (Fig. A7).

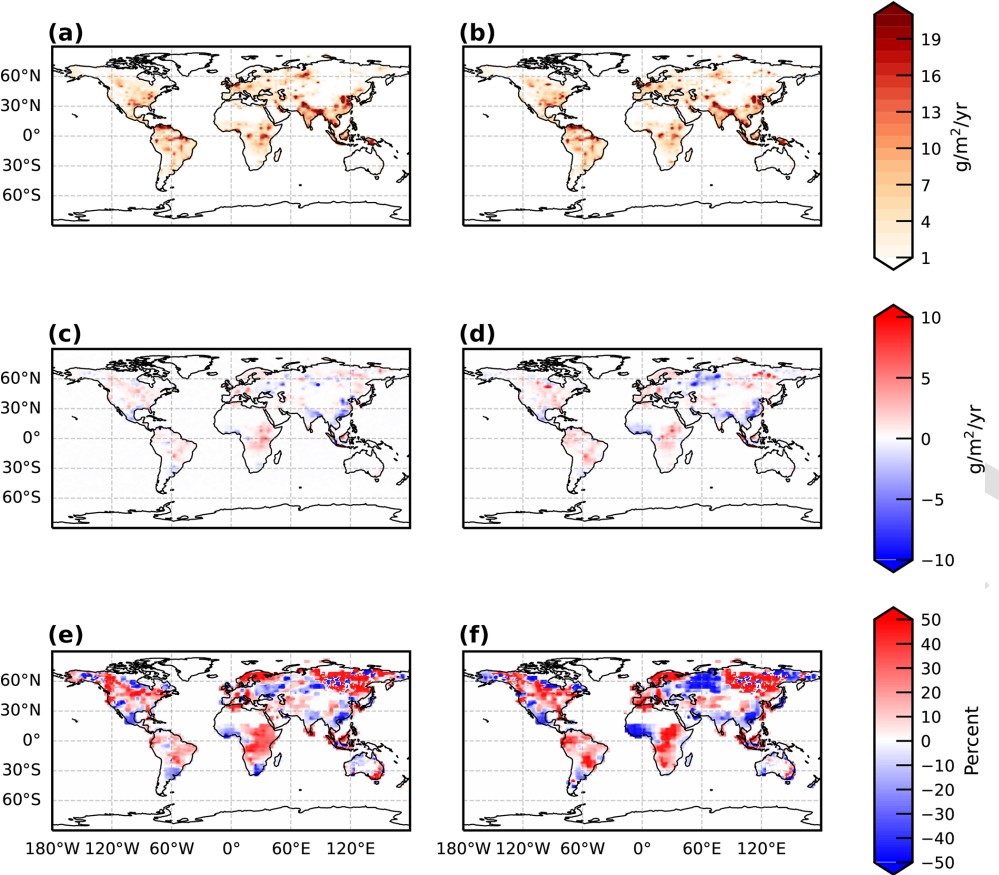

**Figure 2.** Global a posteriori emissions of methane $(\mathrm{g\,m^{-2}\,yr^{-1}})$ inferred from GOSAT methane : $CO_2$ column ratio data for **(a)** 2020 and **(b)** 2021 and how they differ from the baseline year of 2019, described in terms of **(c, d,** respectively) absolute and **(e, f,** respectively) percentage values.

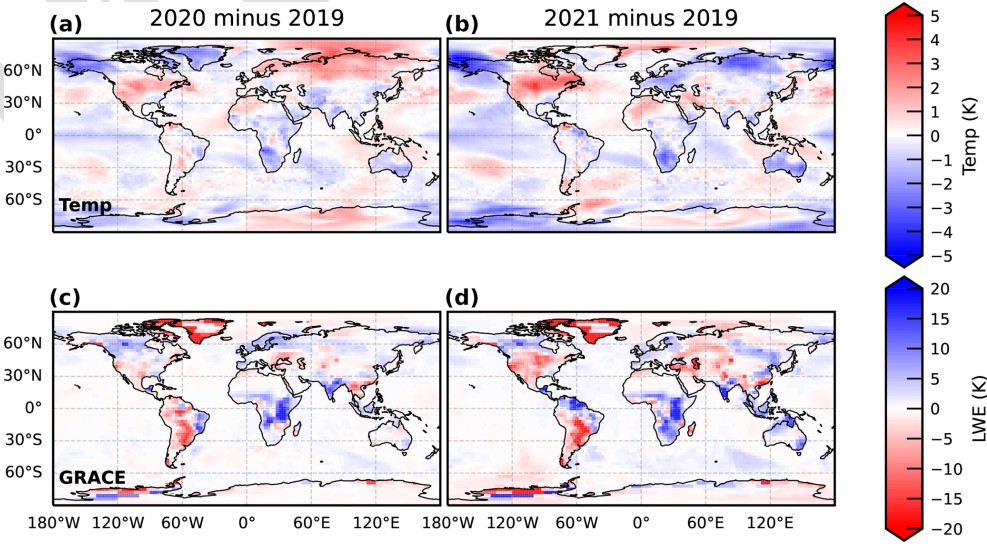

**Figure 3.** Global annual mean surface temperature and GRACE liquid water equivalent (LWE) anomalies in **(a, c)** 2020 and **(b, d)** 2021 relative to values in 2019.

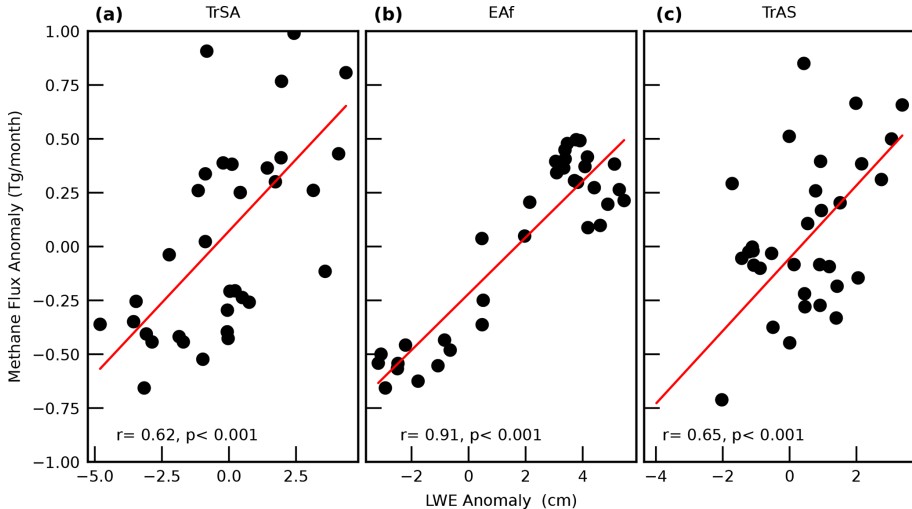

**Figure 4.** Scatter plot of monthly GRACE-FP LWE anomalies (cm) and methane flux anomalies, 2018–2021, over **(a)** tropical South America, **(b)** Eastern Africa, and **(c)** tropical Asia. Red lines denote the linear regression. Numbers atop of each panel denote the Pearson correlation coefficient $r$ and the $p$ value.

By including OH scaling factors into our state vector, we simultaneously infer OH distributions and methane emissions (Sect. 2.4). Table 1 reports the resulting annual changes in tropospheric OH. The annual mean a posteriori OH changes relative to climatological a priori values are $0.91 \pm 1.7\%$, $-0.52 \pm 1.7\%$, and $0.62 \pm 1.7\%$ for 2019, 2020, and 2021, respectively (Table 1). These values correspond to a posteriori OH reductions of 1.43% and 0.29% in 2020 and 2021 relative to the 2019 baseline year (Fig. A3b). Table 1 shows that these reductions in OH correspond to smaller a posteriori methane emissions needed to fit the observations, as expected. For 2019, we estimate a 0.3% higher value for the a posteriori methane emission compared the OH climatology inversion. In 2020, we require an emission increase of 17.9 Tg relative to 2019, an approximate drop of a third in the emission growth needed to reconcile atmospheric observations relative to the OH climatology inversion (Table 1). However, in 2021, we require only a small emission increase of 0.5 Tg (8%) from 2020 due to a concomitant increase in OH compared to a decrease of 6.2 Tg in 2021 for the inversion using OH climatology (Table 1).

Figure 5 shows that annual a posteriori error correlations between the six OH scaling factors and the regional methane emission estimates are only weakly correlated (ranging $\pm 0.1$, and typically less than $\pm 0.05$), suggesting that the GOSAT methane data support the estimation of OH scaling factors on our large geographical scales. Our joint methane–OH inversion results for 2020 are consistent with our simpler OH perturbation studies, described in Sect. 2.3, that are reported in Table 2. For these sensitivity experiments, we find that we need reduced increases in methane emissions in 2020, ranging between $-22.6\%$ and $-27.4\%$. They provide confidence in our a posteriori emission estimates and our

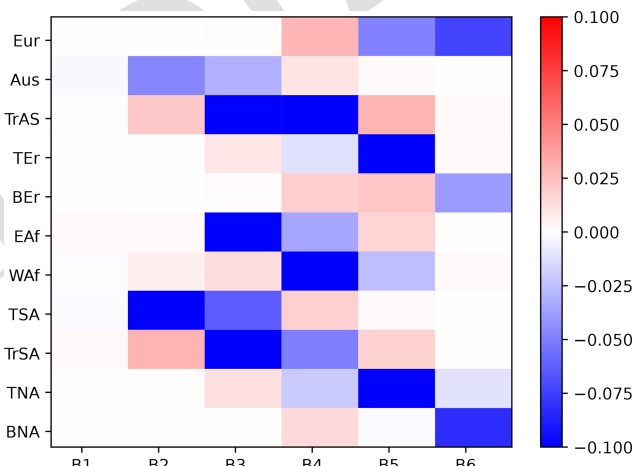

**Figure 5.** A posteriori error correlation between OH scaling factors for six latitude bands (75–50° S, 50–25° S, 25–0° S, 0–25° N, 25–50° N, 50–75° N) and emissions for 13 regions (Fig. 1).

statement that most of the atmospheric methane growth in 2020 was due to an increase in emissions, with the OH reduction being associated with the Covid-19 lockdown representing approximately 30% of the global atmospheric methane growth.

Figures 1c and 6 show the geographical distribution of methane emissions for 2020 and 2021 corresponding to the joint methane–OH inversion and how they differ from a posteriori methane emissions inferred from OH fixed climatology. Figure 7 shows the corresponding a posteriori methane loss due to OH oxidation from the joint methane–OH inversion relative to the baseline inversion that uses OH climatology. Strictly speaking, we cannot easily compare results

**Table 2.** Numerical experiments that explore the influence of assumed OH distributions on a posteriori methane emission estimates.

| Experiment | OH field | 2019–2020 emission increase (Tg yr$^{-1}$) [% difference from control] |
| --- | --- | --- |
| Control run | Fixed OH climatology | $27.0 \pm 7.1$ [–] |
| 5 % OH reduction | Reduction of 5 % OH over regions with high fossil $CO_2$ emission | $20.9 \pm 7.1$ [−22.6] |
| Ozone-like OH reduction | OH reduction following observed ozone change (Ziemke et al., 2022) | $19.6 \pm 7.1$ [−27.4] |
| Joint methane–OH inversion | 18 OH scaling factors for a priori monthly 3-D OH fields inferred from atmospheric methane observations | $20.5 \pm 7.3$ [−24.1] TS1 |

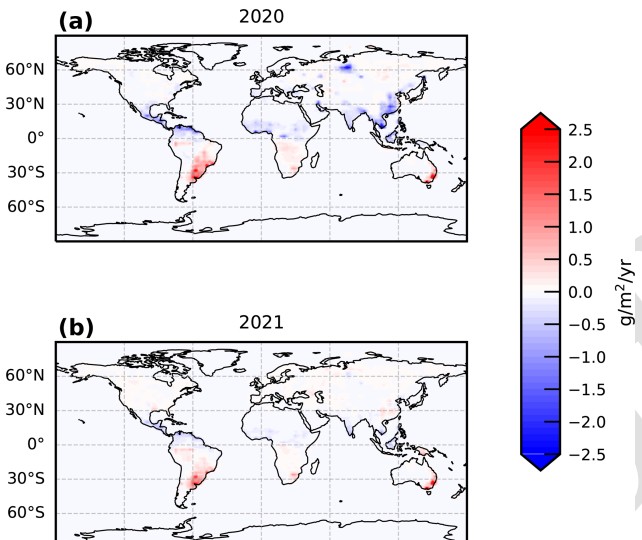

**Figure 6.** Annual mean difference of a posteriori methane emissions between the control inversion that uses OH climatology and the joint OH–flux inversion that estimates OH scaling factors for **(a)** 2020 and **(b)** 2021.

**Figure 7.** Annual mean difference of a posteriori methane loss due to OH oxidation (2020–2021) between the control inversion that uses OH climatology and the joint OH–flux inversion that estimates OH scaling factors.

from the two inversions because the differences are relative to their own pre-2020 baselines (Fig. A3). The 2019 baseline for the joint methane–OH inversion is lower over eastern China, eastern India, and some of boreal Eurasia, and it is higher over parts of tropical and temperate South America and maritime Southeast Asia (Fig. A3). Differences between the inversions are typically much smaller than 10 % of the fluxes. Nevertheless, our main conclusions remain robust for both inversions.

Eastern Africa remains the biggest contributor to atmospheric methane growth in 2020 and 2021 but with the estimated emission increase reduced by $\sim 15$ %, as expected, given the decrease in OH (Fig. 1c). A notable difference includes a large drop in emissions over tropical South America in 2020, but this partially recovers in 2021 (Fig. 1c). The difference between the two inversions for tropical South Amer-

ica in 2020 can be attributed to a decrease in the methane loss due to OH oxidation in the north of that region (Fig. 7) but also the higher 2019 baseline for the joint methane–OH inversion (Fig. A3b) that effectively reduces the emission increase needed to reconcile with observations (Fig. 6). This argument is also relevant to smaller a posteriori methane emissions for the joint methane–OH inversion over Southeast Asia and Southeast Australia (Figs. 1c, 6, A3). A lower 2019 baseline over China and India for the joint methane–OH inversion (Fig. A3) results in small reductions in a posteriori methane emissions needed to reconcile with observations (Figs. 1c, 6). A similar argument associated with a lower 2019 baseline value for the joint methane–OH inversion helps to explain the increase in a posteriori methane emissions over temperate North America and Europe in both years (Figs. 1c, 6).

Our a posteriori emission estimates from our baseline inversion that uses climatological OH values and from joint methane–OH inversion are consistent with independent observations from the TCCON network (Figs. C1 and C2 in Appendix C). A posteriori methane emission estimates for the joint methane–OH inversion generally has slightly smaller differences at southern and northern midlatitude sites (within 1 ppb) but does slightly worse (within 3 ppb) for the Bremen (Br), Karlsruhe (Ka), and Paris sites. Both inversions have comparable standard deviation about the mean differences that are typically within 10 ppb.

## 4 Concluding remarks

We reported regional emission estimates of methane during 2020 and 2021, 2 years with record-breaking atmospheric growth rates, inferred from satellite observations of methane from the Japanese Greenhouse gases Observing SATellite. For our control inversion, we used fixed climatological OH values. Substantial, widespread reductions in nitrogen oxides during 2020 associated with the shutdown of manufacturing and other industries will have perturbed atmospheric concentrations of the OH loss of methane. A reduction in OH could also help explain, in principle, the record-breaking atmospheric increase in methane. To address this point, we also report a posteriori zonal mean OH scaling factors that form part of a joint methane–OH inversion. Generally, we find that our results from the joint methane–OH inversion are broadly consistent with our idealized sensitivity calculations that describe changes in OH informed by distributions in anthropogenic emission and the response of tropospheric ozone, resulting in a reduced emission growth of $17.9\,\mathrm{Tg\,yr^{-1}}$ in 2020 that represents 66 % of our baseline inversion.

We find that emissions from Eastern Africa, tropical South America, and temperate North America play a significant role in determining the global atmospheric growth rate of methane in one or both years. The contribution from Eastern Africa dominates the global growth and is consistent with previous studies that have reported emissions from recent years (Feng et al., 2022b; Pandey et al., 2021; Lunt et al., 2021b, 2019), ranging from 13 to 14 Tg in 2020 and 2021, relative to the 2019 baseline year. The magnitude of this increase in regional emission is reduced by approximately $\sim 15$ % due to a 4 % reduction in OH, as expected. The influence of tropical South America and temperate North America is sensitive to the OH distribution, particularly during 2020. The joint methane–OH inversion results in a substantial decrease in emissions over tropical South America, but this largely recovers by 2021. We find that the joint methane–OH inversion leads to an increase in northern midlatitude methane emissions from temperate North America, temperate Eurasia, and Europe due to a 0.3 % increase in OH. We also find that adjusting OH results in smaller emission reductions from China in 2020 and 2021 and a smaller increase in

emissions from Southeast Asia. We find statistically significant positive correlations between tropical methane emission and hydrological anomalies, consistent with recent studies that have highlighted a growing role for microbial sources over the tropics (Lunt et al., 2019; Feng et al., 2022b; Wilson et al., 2021).

Our results are broadly consistent with a recent study of the 2020 period (Qu et al., 2022), including the magnitude of change associated with a change in OH, albeit concluded using an independent method. Recent work that used in situ methane data reported a smaller increase in methane emissions during 2020 (Peng et al., 2022), consistent with poor data coverage over the tropics as we explain in Appendix B. Their estimate for the Covid-19-related OH reduction during 2020 (1.6 %) is consistent with our estimate, but because they only used surface data in their inversion, they underestimated the atmospheric methane growth from 2019 to 2020 (Table 1) and consequently overestimated the influence of reduced OH on the atmospheric growth rate of methane during 2020.

Our study highlights the tremendous value of using satellite observations to understand rapid changes in atmospheric methane. They provide crucial information to not only identify regional column hotspots associated with emissions but also provide correlative information to help attribute those hotspots to specific anthropogenic or natural emissions. Our study also illustrates the importance of simultaneously estimating methane emissions and changes in OH to improve quantitative knowledge of changes in methane emissions, which is necessary to attribute global atmospheric growth to individual source regions.

## Appendix A: Additional figures

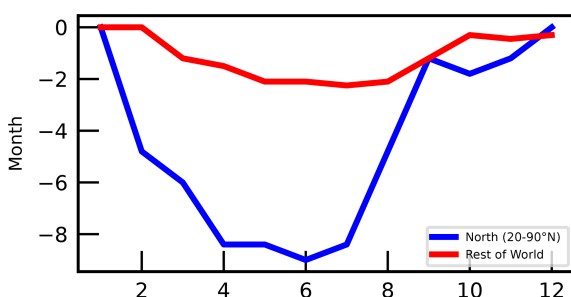

**Figure A1.** Assumed temporal distribution for OH reduction (%) in the Northern Hemisphere (20–90° N, blue line) and the rest of the world (red line) for our sensitivity calculation.

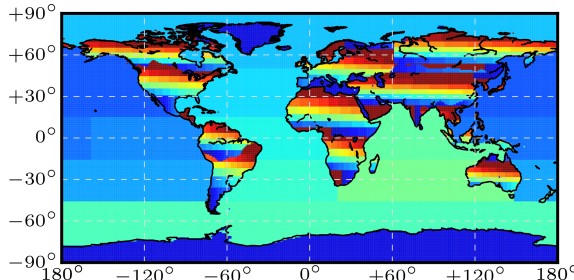

**Figure A2.** Basis functions that describe the 487 regions where we estimate methane emissions, including 476 land regions and 11 oceanic regions.

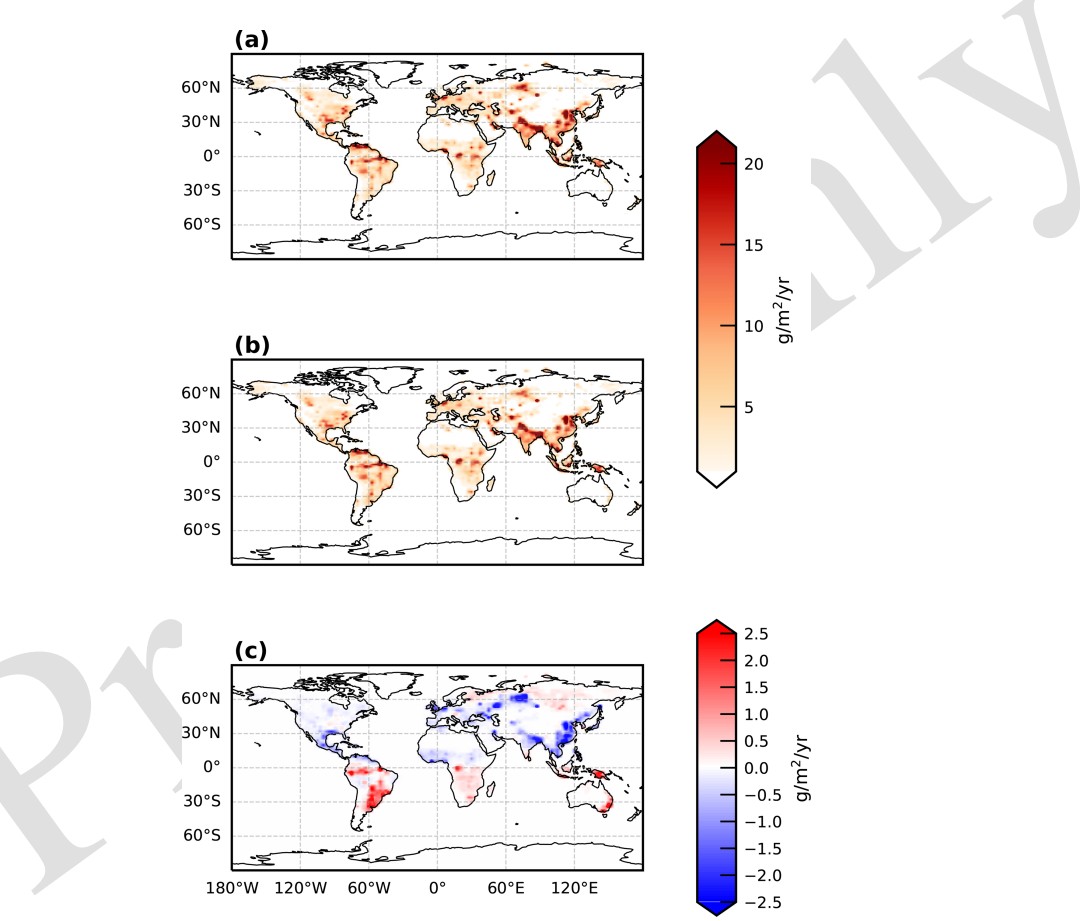

**Figure A3.** Global a posteriori emissions of methane ($\mathrm{g\,m^{-2}\,yr^{-1}}$) inferred from GOSAT methane : $CO_2$ column ratio data for the baseline year of 2019, corresponding to **(a)** OH climatology (Feng et al., 2022b) and **(b)** the joint methane–OH inversion. Panel **(c)** shows the difference of a posteriori emissions of methane corresponding to the joint methane–OH inversion minus OH climatology.

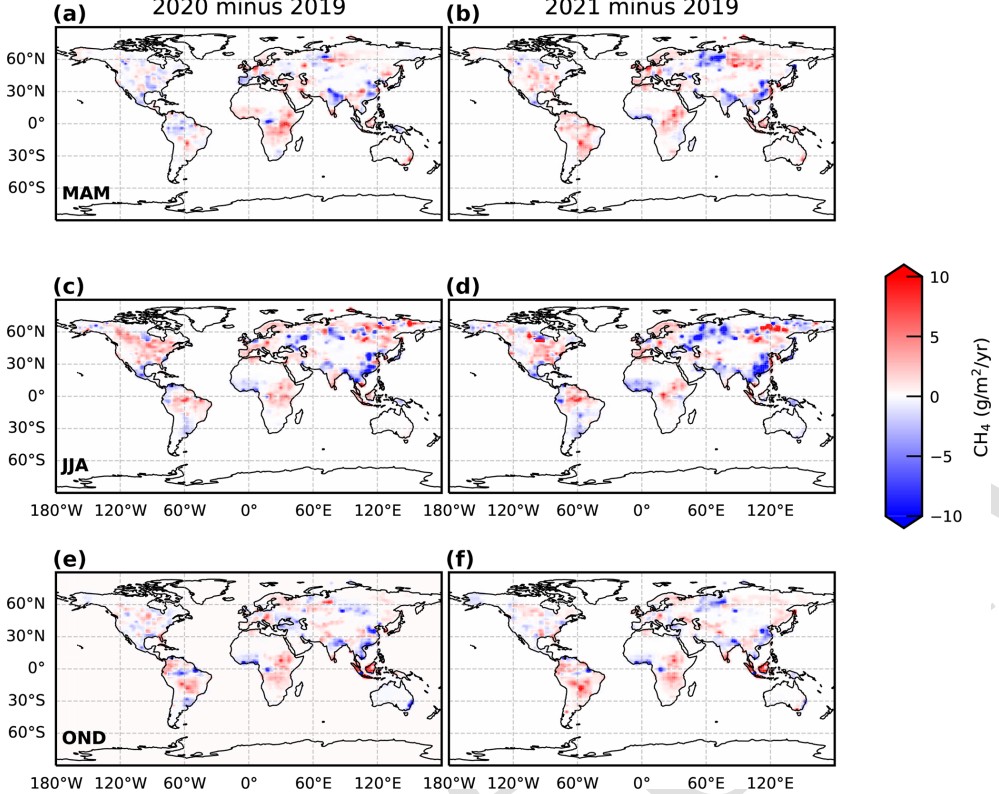

**Figure A4.** Global seasonal a posteriori emissions of methane $(g\,m^{-2}\,yr^{-1})$ inferred from GOSAT methane : $CO_2$ column ratio data for **(a, c, e)** 2020 and **(b, d, f)** 2021 relative to the baseline year of 2019, described in terms of absolute values. Seasons are based on rainfall changes over the tropics.

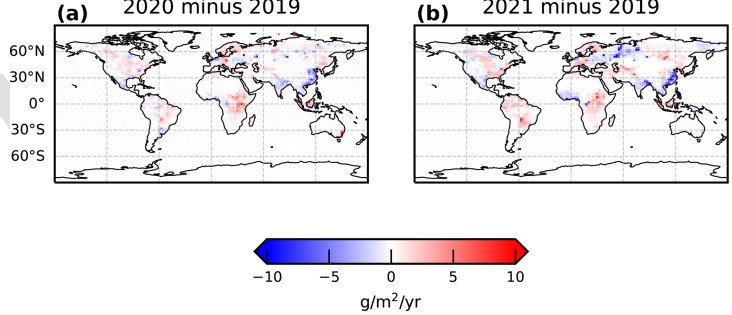

**Figure A5.** Global a posteriori emissions of methane $(g\,m^{-2}\,yr^{-1})$ inferred from GOSAT methane column data for **(a)** 2020 and **(b)** 2021.

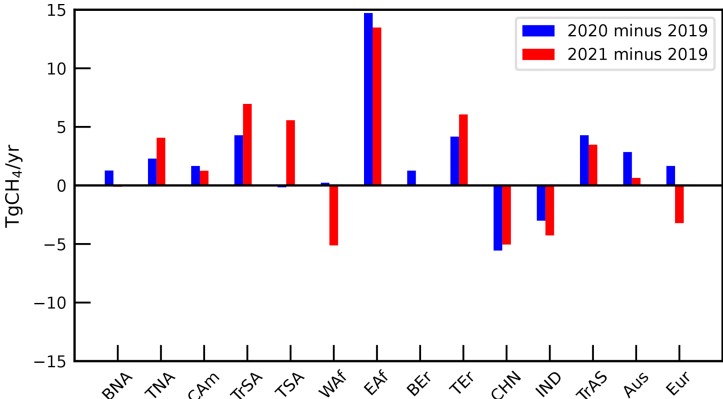

**Figure A6.** The same as Fig. 1b but for an inversion that uses GOSAT proxy XCH$_4$ and in situ methane data.

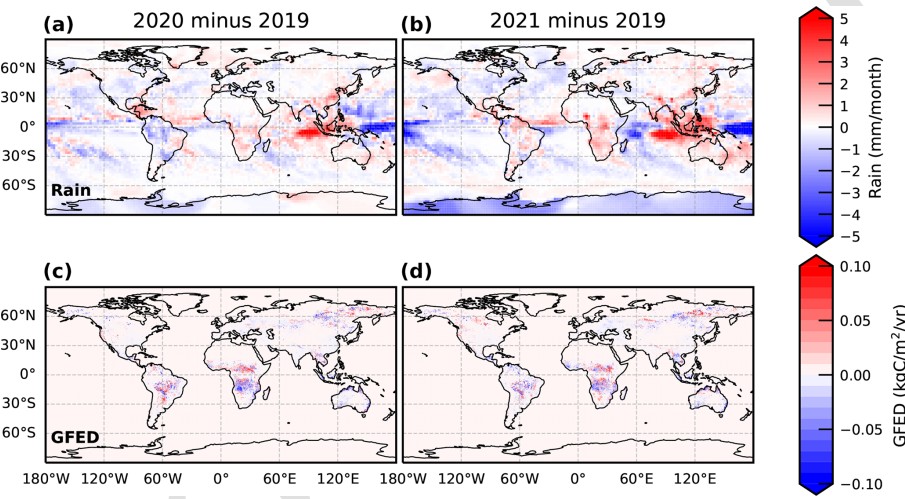

**Figure A7.** Global annual mean NOAA CMAP precipitation (mm month$^{-1}$ yr$^{-1}$) and GFED fire emission (kgC m$^{-2}$ yr$^{-1}$) anomalies in **(a, c)** 2020 and **(b, d)** 2021 relative to values in 2019.

## Appendix B: Description of box model calculation

To calculate global emissions of methane from the NOAA global mean data, we use a simple one-box model. In this model, the change in global mean methane concentration over time is given by TS2

$$\frac{\mathrm{d}B}{\mathrm{d}t} = Q - kB,$$

where $B$ is the atmospheric mass of methane in Tg, $k$ is the loss rate given as 1 per (unit) lifetime CE3 and $Q$ is the emissions rate. From this, after integration, the annual emissions rate can be calculated as follows:

$$Q_t = \frac{k(B_t - B_{t-1} \cdot e^{-k})}{(1 - e^{-k})}.$$

The loss rate was tuned to match a steady state concentration of 1775 ppb during 2000–2006 based on constant emissions of 530 Tg yr$^{-1}$ during this period. We calculated the rolling 12-month annual emissions to track the progression of global emissions between 2019 and 2021. We used the difference between the atmospheric concentration in January 2019 and January 2020, February 2019 to February 2020, etc. to calculate the change in emissions in the intervening 12 months. Figure B1 shows the increase in emissions throughout 2020 followed by more variable month-to-month changes in 2021. The large increase in emissions primarily occurs in 2020, with emissions at the 12-month period ending in December 2020 being 27 Tg yr$^{-1}$ larger than the emissions 1 year earlier. In contrast, if emissions are calculated using annual mean concentrations, it appears as if there is a larger emission

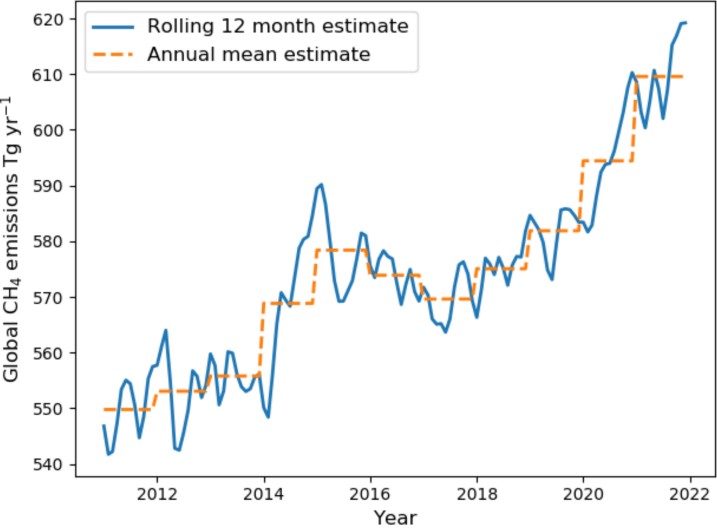

**Figure B1.** Global box model methane emission estimates between 2011 and 2021, respectively. Emission estimates are based on NOAA global mean surface data. The blue line denotes the rolling 12-month annual emissions, and the orange line denotes the emissions based on annual mean concentrations.

increase in 2021. The box model results show that the highly simplified calculation based on global average data is consistent with the more complex inverse modelling approach applied to the GOSAT data.

## Appendix C: Evaluation of a posteriori flux estimates

We indirectly evaluate our a posteriori methane fluxes by comparing the GEOS-Chem methane distribution, driven by the a posteriori fluxes, with independent $XCH_4$ retrievals from the Total Carbon Column Observing Network (TCCON) of Fourier transform spectrometers (Wunch et al., 2022). We use bias-corrected TCCON $XCH_4$ data from the latest GGG2020 public release of the TCCON dataset from 2019 to 2021, including updates until October 2022. For a comprehensive description of the network and the available data from each TCCON site, we refer the reader to the TCCON project page. Here, we use a subset of available TCCON data, dependent on their availability between 2018 and 2021 (Buschmann et al., 2022; De Mazière et al., 2022; García et al., 2022; Hase et al., 2022; Kivi et al., 2022; Liu et al., 2022; Morino et al., 2022a, b, c; Notholt et al., 2022; Petri et al., 2022; Pollard et al., 2022; Warneke et al., 2022; Shiomi et al., 2022; Té et al., 2022; Wennberg et al., 2022a, b, c; Wunch et al., 2022; Zhou et al., 2022). For further details about the data, we direct the reader to the TCCON project page: http://tccondata.org/ (last access: 15 December 2022).

Figure C1 shows the mean and standard deviation of the differences between our a posteriori model simulation and TCCON GGG2020 data in 2019 and 2020. The a posteriori model simulation is driven by our a posteriori methane emission estimates. We sample the associated model 3-D atmospheric methane distributions at the time and location of each TCCON site used. We then convolve the sampled vertical profile with site- and time-dependent TCCON instrument averaging kernels, which describes the altitude-dependent instrument sensitivity to changes in atmospheric methane concentration. Figure C2 shows the same statistical comparison but using the a posteriori methane emission estimates inferred with the OH scaling factors, as described in Sect. 2.4.

We do not report results for 2021 due to data availability. For data that are available, we find that the mean statistics (not shown) are similar to those we report here for 2019 and 2020. For most sites, we find that the mean bias is typically smaller than $\pm 10$ ppb and the standard deviation has a range 5–15 ppb with values typically smaller than 10 ppb. We find the largest differences at northern high latitudes where the model has a large overestimate ($\sim 10$ ppb), consistent with previous studies (Feng et al., 2017, 2022b), due to poor coverage of GOSAT data during the boreal winter and to model error.

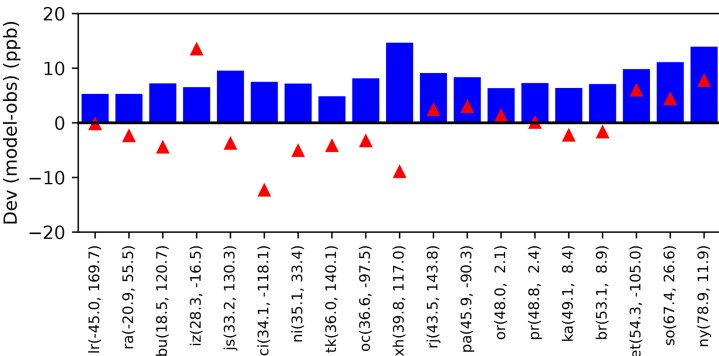

**Figure C1.** Statistical comparison of the GEOS-Chem a posteriori methane distribution and TCCON XCH$_4$ data (v GGG2020) in 2019 and 2020. Red upward triangles denote the mean bias and the blue bars denote the corresponding 1s values.

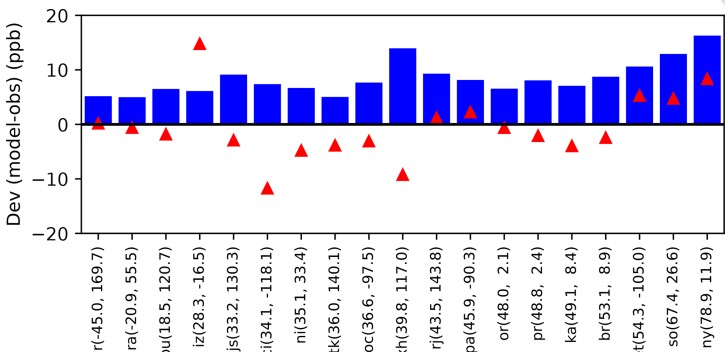

**Figure C2.** The same as Fig. C1 but for the inversion that also includes OH scaling factors.

**Data availability.** The University of Leicester GOSAT Proxy v9.0 XCH$_4$ data are available from the data repository of the Centre for Environmental Data Analysis at https://doi.org/10.5285/18ef8247f52a4cb6a14013f8235cc1eb (Parker and Boesch, 2020). Precipitation, temperature, and the GRACE datasets are available at https://doi.org/10.5067/5ESKGQTZG7FO (GMAO, 2015) and https://doi.org/10.5067/TEMSC-3MJC6 (Wiese et al., 2018) TS3. The community-led GEOS-Chem model of atmospheric chemistry and transport model is maintained centrally by Harvard University (http://geos-chem.seas.harvard.edu, last access: 8 March 2021), and is available on request. The ensemble Kalman filter code is publicly available as PyOSSE (https://www.nceo.ac.uk/data-tools/atmospheric-tools/, NCEO, 2023 TS4). The TCCON data were obtained from the TCCON Data Archive hosted by CaltechDATA at https://doi.org/10.14291/TCCON.GGG2020 (TCCON Team, 2022) TS5. The CH$_4$ GLOBALVIEWplus v5.0 ObsPack is available from https://doi.org/10.25925/20221001 (Di Sarra et al., 2022b), and the CO$_2$ GLOBALVIEWplus v8.0 ObsPack is available from https://doi.org/10.25925/20220808 (Cox et al., 2022a).

**Author contributions.** LF and PIP designed the research; LF and MFL prepared the calculations; RJP and HB provided the GOSAT data and expert advice on its usage; PIP wrote the paper, with comments from LF, RJP, MFL, and HB.

**Competing interests.** The contact author has declared that none of the authors has any competing interests.

**Disclaimer.** Publisher's note: Copernicus Publications remains neutral with regard to jurisdictional claims in published maps and institutional affiliations.

**Acknowledgements.** We thank the Japanese Aerospace Exploration Agency, National Institute for Environmental Studies, and the Ministry of Environment for the GOSAT data for their continued support as part of the Joint Research Agreements at the Universities of Edinburgh and Leicester. GOSAT retrievals were processed using the ALICE High-Performance Computing Facility at the University of Leicester. We thank all the scientists that submitted data to the CO$_2$ and methane Observation Package (ObsPack) data products, coordinated by NOAA ESRL, and making them freely available for

carbon cycle research. We also thank the GEOS-Chem community, particularly the team at Harvard University who helped to maintain the GEOS-Chem model, and the NASA Global Modeling and Assimilation Office (GMAO) who provided the MERRA2 data product.

**Financial support.** Liang Feng, Paul I. Palmer, Robert J. Parker, and Hartmut Bösch received support from the UK National Centre for Earth Observation funded by the National Environment Research Council (grant no. NE/R016518/1); Robert J. Parker also received funding from grant no. NE/N018079/1. We received funding from the Copernicus Climate Change Service (C3S2_312a_Lot2) related to the generation of the GOSAT data.

**Review statement.** This paper was edited by Bryan N. Duncan and reviewed by two anonymous referees.

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

## Remarks from the language copy-editor

## Remarks from the typesetter