# Peer review of "Methane emissions are predominantly responsible for recordbreaking atmospheric methane growth rates in 2020 and 2021"

_Atmospheric Chemistry and Physics, 2022_

## Referee Comment (RC1)

**1 Overview:**

Review of "*Methane emissions responsible for record-breaking atmospheric methane growth rates in 2020 and 2021*" by Feng *et al.*

Feng *et al.* present a brief analysis of a set of methane inversions for 2019-2022 using an EnKF and GOSAT observations. They find spatial changes in methane emissions and look at correlative data such as GRACE. They conduct two sensitivity studies using different prescribed OH levels. The authors conclude that methane emissions are responsible for the fast methane growth observed in 2020 and 2021. A crucial aspect of this work that is missing is the evaluation. There seems to be no evaluation of the results using independent observations or techniques like $k$-fold cross validation. The work addresses an important and timely topic but, in this reviewer's opinion, the main claims in the manuscript (that the growth is driven by emissions, not chemistry) do not seem supported by their numerical experiments. This reviewer would recommend major revisions to evaluate their results and either reframe what is actually being concluded or provide evidence supporting their conclusions.

**2 Review Criteria:**

1. Does the paper address relevant scientific questions within the scope of ACP? **Yes.**
2. Does the paper present novel concepts, ideas, tools, or data? **Yes.**
3. Are substantial conclusions reached? **Yes.**
4. Are the scientific methods and assumptions valid and clearly outlined? **Mostly.**
5. Are the results sufficient to support the interpretations and conclusions? **No.**
6. Is the description of experiments and calculations sufficiently complete and precise to allow their reproduction by fellow scientists (traceability of results)? **No.**
7. Do the authors give proper credit to related work and clearly indicate their own new/original contribution? **No. Some important references are missing.**
8. Does the title clearly reflect the contents of the paper? **Yes.**
9. Does the abstract provide a concise and complete summary? **Yes.**
10. Is the overall presentation well structured and clear? **Yes.**
11. Is the language fluent and precise? **Yes.**
12. Are mathematical formulae, symbols, abbreviations, and units correctly defined and used? **Yes.**
13. Should any parts of the paper (text, formulae, figures, tables) be clarified, reduced, combined, or eliminated? **No.**
14. Are the number and quality of references appropriate? **No. Missing some important references.**
15. Is the amount and quality of supplementary material appropriate? **N/A. No supplement. This reviewer feels that some of the appendices should be brought into the main text.**

**3 Major Comments:**

**3.1 Robustness of attribution to emissions or chemistry**

The major concern this reviewer has with the manuscript is that the title and central claims don't seem supported by their data. The main scientific claim (and their final conclusion) is that the record-breaking methane growth rates in 2020 and 2021 were driven by emissions, not chemistry. This claim certainly seems plausible (if not likely), but their experiments do not seem sufficient to justify that claim.

In this reviewer's opinion, the ideal way to conclude as to the relative importance of emissions and chemistry would be to include both emissions and OH in the state vector for their EnKF. That would result provide a straight forward assessment of the relative role of each process. The argument presented in this manuscript, as this reviewer interpreted it, is as follows:

- the authors conducted a global inversion at $2° \times 2.5°$ resolution with an EnKF from 2019-2022. This inversion assumes constant OH fields for the 3-year window. The authors find changes in the magnitude and spatial patterns of methane emissions.

- the authors compared these emission changes to rainfall, GRACE groundwater, and temperature. The largest correlations were 0.5-0.6 (representing 25–35% of the variability).

- the authors conducted a second global inversion with the same setup but reduced OH by 5% where the largest COVID changes occurred.

The authors show the difference in emissions resulting from these cases but it is not clear to this reviewer which result is better. The differences seem to be central to their conclusions as indicated in the last two lines of their abstract (*"Based on a sensitivity study for which we assume a conservative 5% decrease in hydroxyl concentrations in 2020...we find that the global increase in our a posteriori emissions in 2020 is ∼22% lower than our control calculation. We conclude therefore that most of the observed increase in atmospheric methane during 2020 and 2021 is due to increased emissions."*) but I could not discern how they concluded why one was better than the other. Specifically, it is unclear why the control calculation is the correct answer here.

**3.1.1 Evaluation and/or overfitting?**

Two common methods for evaluating the performance of optimization schemes are to: 1) evaluate against independent observations or 2) perform $k$-fold cross validation. Neither of these were included here. This is something that should be included for all their cases with an inversion analysis to ensure that one is not overfitting for a particular inversion.

**3.1.2 OH is inconsistent with other work**

The authors chose a 5% reduction in OH based on Laughner et al. (2021). However, Laughner et al. (2021) was a review/synthesis paper that took global mean OH changes

from Miyazaki et al. (2021; doi:10.1126/sciadv.abf7460) and used them in a box model.

This reviewer wonders how large the *global* mean OH changes are in this manuscript from Feng et al.? My suspicion is that they are quite a bit smaller than what was reported in Miyazaki et al. and Laughner et al.

Additionally, the OH chemistry is highly non-linear and Miyazaki et al. discuss how OH and ozone actually increase in some regions despite the $NO_x$ reductions. Using OH fields from Miyazaki et al. would be a much better way of testing if the OH simulated in that work impacted the methane burden.

Essentially, this reviewer does not think the OH sensitivity run designed here accurately portrays the OH changes that others have found. Data supporting the choice of OH runs used here would help assuage these concerns.

**3.2 GOSAT proxy observations**

The authors use GOSAT proxy observations. This means that the methane concentrations will be dependent on the $CO_2$ concentrations. However, it seems like the authors use $CO_2$ simulations with monthly emissions through 2019. Therefore the $CO_2$ could lead to a bias in their methane concentrations during COVID due to the reduction in $CO_2$ emissions. This would be most pronounced in urban areas.

This reviewer was also very confused by the description of the data used in places. For example, when describing a sensitivity study the authors mention using proxy GOSAT $XCH_4$ data (Line 90) but the main inversions also seem to use proxy GOSAT data.

**4 Minor Comments:**

**4.1 Oversight of previous work**

The authors seem to have overlooked important recent literature on this topic including, for example, McNorton et al. (2022; doi:10.5194/acp-22-5961-2022) who used TROPOMI data to constrain methane emissions during COVID.

**4.2 Uncertainties**

The authors don't seem to have reported uncertainties. It's clear what changes are actually substantial or within the noise. For example, the abstract lists changes of -3 Tg and -5 Tg as "substantial" in the abstract (Line 18). These don't seem particularly large. The text later claims that their work is within the uncertainty of another paper (Line 119), so it would be good to see uncertainties reported throughout.

**4.3 Error correlations**

Where do the temporal and spatial prior error correlations come from? It seems that the authors use spatial correlation lengths of 300 km and 1 month. Are these important in the spatial patterns found here?

**4.4 Introduction**

This reviewer is a bit confused by the list of citations in the intro. Specifically, Lines 34-35. The authors claim there is an intense debate on the role of fast growth in 2020 and 2021. They then claim that work has shown the importance of regional anomalies in the tropics. But many of these studies are from earlier than the time period being discussed.

"The underlying reasons for these anomalous growth rates in 2020 and 2021 are currently subject to intense debate with some studies attributing most of the growth in 2020 to a reduction in the hydroxyl radical (OH) sink of methane due to global-scale reductions in nitrogen oxides due to pandemic-related industry shutdowns (Laughner et al. 2021). On the face of it, this appears to be a reasonable explanation, but recent studies have used satellite observations of atmospheric methane to reveal regional hotspots over the tropics that are responding to changes in climate and have global significance (Pandey et al. 2021; Lunt et al. 2019; 2021; Pandey et al. 2017; Feng et al. 2022; Palmer et al. 2021; Wilson et al. 2020)."

**4.5 Correlative data**

The authors show plots of the changes in correlative data, but don't show spatial correlations. In this reviewer's opinion, it would be helpful to show a map with the correlation between the emission anomalies and the correlative data. The manuscript currently requires the reader to make the connection themself.

---

## Referee Comment (RC3)

[revised manuscript text omitted]

**2-1** — Aug 5, 2022 at 3:00 PM, Reviewer

By how much?

**3-1** — Aug 5, 2022 at 3:00 PM, Reviewer

For both CO2 and CH4, or just CH4?

**3-2** — Jul 1, 2022 at 4:53 PM, Reviewer

Olsen and Randerson is a downscaling technique. What biospheric model did you use?

**3-3** — Aug 5, 2022 at 3:00 PM, Reviewer

A 5% reduction is not that conservative. The spatial pattern of OH reduction is quite important. Miyazaki et al, 2021 (Science Advances) shows that the global mean reduction in OH is about 4%. However, the reductions can be substantially higher, (>30%) locally. These will directly affect the methane inferences, especially over Europe and parts of Asia.

Please review the paper and perform a sensitivity analysis closer to reported spatial changes in OH.

**3-4** — Aug 5, 2022 at 3:00 PM, Reviewer

We could expect, however, that the CO2 emission in 2020 and 2021 to be different than 2019. Not sure how this really addresses the OH issue.

As noted next, the Miyazaki et al, 2021 provides a more observationally constrained pattern of OH.

**3-5** — Jul 1, 2022 at 4:53 PM, Reviewer

Is this a correlation in flux or concentration space? If it is the former, where did 300km come from?

**4-1** — Jul 1, 2022 at 4:53 PM, Reviewer

You should be able to calculate the simulated atmospheric CH4 growth rate for 2020 and 2021 from your inversion. You should add that to the table (1).

The box-model approach carries its own assumptions, and can't be really used to validate the CH4 top-down estimates.

**4-2** — Jul 1, 2022 at 4:53 PM, Reviewer

Based upon NOx emission changes during 2020-2021, we would not expect CH4 lifetime to be fixed.

**5-1** — Aug 5, 2022 at 3:00 PM, Reviewer

There are no uncertainty estimates in this calculation. I can not assess whether these changes are statistically significant or not.

Uncertainty estimates need to be added to these flux estimates. These could be done through comparison with other inversions or independent observations.

**5-2** — Jul 1, 2022 at 4:53 PM, Reviewer

It looks like the increases in SA in 2021 are an acceleration of the 2020 increases. What is going on there?

**5-3** — Jul 1, 2022 at 4:53 PM, Reviewer

How is anomalous defined here? The temperature variations need to be calculated relative to the long term temperature variability to assess whether they are not within climatology.

**5-4** — Jul 1, 2022 at 4:53 PM, Reviewer

Please put plots of those regressions in the supplemental.

**6-1** Aug 5, 2022 at 3:00 PM, Reviewer

Cooper didn't discuss NOx emissions, only concentrations. The impact on OH would need to be calculated separately and would be affected by other reactive species, e.g., ozone.

**6-2** Jul 1, 2022 at 4:53 PM, Reviewer

It's hard to see the differences in a spatial plot. Please make a difference plot using the regions used in Fig. For 2020 and 2021 between the proxy retrieval and the simultaneous estimate.

**6-3** Aug 5, 2022 at 3:00 PM, Reviewer

The implication of East Africa as the single largest driver of the methane growth rate is puzzling. Yes, there is a correlation with water anomalies, but it's not the largest driver of methane in driver.

A more plausible set of physical mechanisms needs to be proposed for why we could expect this region as a dominant driver.

**7-1** Aug 5, 2022 at 3:00 PM, Reviewer

This is not clear to me at all that this is a valid assumption. The OH responses will depend on the regional NOx emission reductions and the timing of those reductions, in part because it will also affect ozone production. There are better sources for OH than this crude assumption.

---

## Author Comment (AC1)

**Response to review comment 1**

We thank the reviewer for their feedback. We respond below to each individual comment raised by this reviewer.

*A crucial aspect of this work that is missing is the evaluation. There seems to be no evaluation of the results using independent observations or techniques like k-fold cross validation. The work addresses an important and timely topic but, in this reviewer's opinion, the main claims in the manuscript (that the growth is driven by emissions, not chemistry) do not seem supported by their numerical experiments.*

We have indirectly evaluated our results for 2020 by inferring methane emissions using the NOAA surface data, which provide consistent results on global and continental spatial scales. We will make this point clearer in the revised manuscript. At this time we do not have access to NOAA data for 2021.

We have also evaluated our posterior emissions using independent TCCON XCH4 measurements for 2020 and the first half of 2021 (not shown). We find that the annual mean model minus TCCON XCH4 values for 2020 are within 10 ppb for 15 (out of 18) sites with standard deviation between 5 and 15 ppb but typically < 10ppb. This is consistent with our evaluation of data from 2009-2019 (Feng et al, 2022), building on previous studies that have included substantial evaluation using a range of in situ and remote sensing data. Our preliminary analysis of an incomplete year of TCCON data in 2021 shows similar model performance. We will accordingly update the text.

The suggestion to use k-fold cross validation is interesting. We have used this method in other less computationally-intensive applications but to our knowledge it has not been used to evaluate a global inversion. We are unconvinced this will add much to our narrative, especially given the focus of the review is on 2020.

*The major concern this reviewer has with the manuscript is that the title and central claims don't seem supported by their data. The main scientific claim (and their final conclusion) is that the record-breaking methane growth rates in 2020 and 2021 were driven by emissions, not chemistry. This claim certainly seems plausible (if not likely), but their experiments do not seem sufficient to justify that claim. In this reviewer's opinion, the ideal way to conclude as to the relative importance of emissions and chemistry would be to include both emissions and OH in the state vector for their EnKF. That would result provide a straight forward assessment of the relative role of each process.*

An alternative way would indeed be to include emissions of methane and OH in the state vector vector but as the reviewer will be aware there are also gross assumptions associated with this approach and the resulting posterior results would not provide a "straightforward assessment" due to the limited information content of the satellite and *in situ* atmospheric methane data. Inversions that also include the estimation of OH, quantify OH values on global or hemispheric scales and therefore will be unable to identify OH hotspots. The ideal approach would be to include the methane emissions in a state vector as part of a full-chemistry data assimilation system that takes into account more appropriate constraints on OH. But of course this problem becomes progressively more complex (non-linear) and more intractable (associated with more chemistry constituents and more data).

The approach we used to quantify the impact of OH is admittedly a brute-force approach but in the absence of rigorous constraints on OH concentrations we feel this is a transparent approach that is

easy to understand. We will include a discussion of this caveat in the revised manuscript. See also our response below, which discusses larger OH perturbations.

*The argument presented in this manuscript, as this reviewer interpreted it, is as follows:*

*the authors conducted a global inversion at 2◦ × 2.5◦ resolution with an EnKF from 2019-2022. This inversion assumes constant OH fields for the 3-year window. The authors find changes in the magnitude and spatial patterns of methane emissions. the authors compared these emission changes to rainfall, GRACE groundwater, and temperature. The largest correlations were 0.5-0.6 (representing 25–35% of the variability). the authors conducted a second global inversion with the same setup but reduced OH by 5% where the largest COVID changes occurred.*

We are certainly more confident in the geographical distribution of emissions rather than their attribution. A more detailed study about the attribution of those emissions will be forthcoming but it is outside the scope of this study.

Saying that, the spatial and seasonal variations in tropical methane emissions are consistent with those reported for earlier years (e.g., Lunt et al, 2019, 2021; Feng et al 2022; Wilson et al, 2021) that showed the response of methane emissions was consistent with microbial sources. However, we think imperfect attribution does not detract from our key message: large emissions are predominantly responsible for anomalous atmospheric growth rates in 2020 and 2021.

*The authors show the difference in emissions resulting from these cases but it is not clear to this reviewer which result is better. The differences seem to be central to their conclusions as indicated in the last two lines of their abstract ("Based on a sensitivity study for which we assume a conservative 5% decrease in hydroxyl concentrations in 2020...we find that the global increase in our a posteriori emissions in 2020 is ~22% lower than our control calculation. We conclude therefore that most of the observed increase in atmospheric methane during 2020 and 2021 is due to increased emissions.") but I could not discern how they concluded why one was better than the other. Specifically, it is unclear why the control calculation is the correct answer here.*

We did not conclude one was better than the other. We concluded that a 5% drop in OH was too large using our perturbation approach, which described only 22% of the emission from our control run. Consequently, most of the atmospheric growth in 2020 and 2021 was due to emissions. We will make this point clearer in the revised manuscript. We will also put this discussion in the context of higher, localized reductions in OH as raised as a discussion point by this reviewer (see below).

*Evaluation and/or overfitting*

*Two common methods for evaluating the performance of optimization schemes are to: 1) evaluate against independent observations or 2) perform k-fold cross validation. Neither of these were included here. This is something that should be included for all their cases with an inversion analysis to ensure that one is not overfitting for a particular inversion*

This study is a two-year extension of recent work (Feng et al, 2022), including substantial evaluation of the results that build on previous studies. In an earlier response (see above) we have described our evaluation with TCCON data for 2020. We have also indirectly evaluated our 2020 results using an inversion constrained by NOAA data that leads to consistent results.

Since we use an ensemble Kalman filter the error characterization is a natural extension of our analysis. Using the cost function weights, we find no evidence to suggest we have overfitted the data. We will add text to that effect in the revised manuscript.

*OH is inconsistent with other work*

*The authors chose a 5% reduction in OH based on Laughner et al. (2021). However, Laughner et al. (2021) was a review/synthesis paper that took global mean OH changes from Miyazaki et al. (2021; doi:10.1126/sciadv.abf7460) and used them in a box model. This reviewer wonders how large the global mean OH changes are in this manuscript from Feng et al.? My suspicion is that they are quite a bit smaller than what was reported in Miyazaki et al. and Laughner et al Additionally, the OH chemistry is highly non-linear and Miyazaki et al. discuss how OH and ozone actually increase in some regions despite the NOx reductions. Using OH fields from Miyazaki et al. would be a much better way of testing if the OH simulated in that work impacted the methane burden. Essentially, this reviewer does not think the OH sensitivity run designed here accu- rately portrays the OH changes that others have found. Data supporting the choice of OH runs used here would help assuage these concerns.*

Our sensitivity experiments are designed to examine how large-scale reductions in OH (associated with Covid-19 lockdowns) affects our methane emission estimates inferred from atmospheric methane measurements.

First, it was remiss of us not to include Miyazaki et al, 2021 for which we apologize. This primary reference will be added to the revised manuscript. Figure S8A from this paper shows localized OH reductions of 15-20% for May 2020, although the authors note that this reduction can be as large as 30% on a grid basis. This represents one of the early months of the shutdown when we expect the largest reduction in emissions so we expect global mean annual changes to be smaller. We also note that  Miyazaki et al, 2021 (as with others) do not consider the coincident changes in non-methane hydrocarbons (NMHCs) that will also affect non-linear ozone chemistry, including the production and loss of OH. Large coincident reductions in formaldehyde, for instance, a proxy of NMHCs, have since been reported in other studies (e.g. Sun et al, 2021). Since publishing his 2021 paper, Dr Miyazaki has re-run his calculations with coincident changes in hydrocarbons (private communication)  and we are currently discussing how this information could potentially inform our results.

We did a number of simulation runs in preparation for this manuscript. We used 1% and 5% global reductions in OH but penalized regions that would not have been impacted by reductions in nitrogen oxide emissions from Covid. We instead chose to reduce OH only over regions where there are substantial anthropogenic $CO_2$ emissions, resulting in a distribution similar to Miyazaki et al, 2021. An effective approach to identify regions affected by the shutdowns. In response to this reviewer comment, we prepared a calculation in which we decrease OH by an additional 25% of eastern China for March 2020 (the peak of emission reductions). We find that this month-long perturbation can explain an additional 0.26 Tg of methane emissions for 2020, providing some indication of how a larger, localized reduction in OH will impact global changes in methane emission estimates. We will add a broader discussion of this kind of OH perturbation to the revised manuscript.

While we are happy to explore different OH perturbations, we hasten to add that this reviewer comment should be tempered by the uncertainties associated with previous studies that do not accurately describe the photochemical perturbation associated with the Covid-19 shutdowns. We will add text that describes caveats associated with both approaches.

*GOSAT proxy observations*

*The authors use GOSAT proxy observations. This means that the methane concentrations will be dependent on the $CO_2$ concentrations. However, it seems like the authors use $CO_2$ simulations with monthly emissions through 2019. Therefore the $CO_2$ could lead to a bias in their methane concentrations during COVID due to the reduction in $CO_2$ emissions. This would be most pronounced in urban areas.*

As described in Parker et al., 2020, the GOSAT data use an ensemble of CO2 models based on atmospheric *in situ* inversions. For recent years, when updated model inversions based on *in situ* observations are not typically readily available on the timescales required for data processing, we have used CO2 values from previous years that have been incremented by the NOAA global growth rate. To ensure that this does not introduce an error due to model CO2 values, an inversion is performed that directly uses the XCH4/XCO2 ratio, without relying (explicitly or implicitly) on the model CO2. We will clarify this point in the revised manuscript.

*This reviewer was also very confused by the description of the data used in places. For example, when describing a sensitivity study the authors mention using proxy GOSAT $XCH_4$ data (Line 90) but the main inversions also seem to use proxy GOSAT data.*

We use two approaches: 1) the proxy ratio (XCH4/XCO2) directly without need to assume a prior model for XCO2 to multiply out to obtain XCH4, which is described in Feng et al, 2022; and 2) the proxy methane XCH4. We find both approaches lead to consistent results. We will amend the text to make this point clearer.

**4 Minor Comments:**

**4.1 Oversight of previous work**

*The authors seem to have overlooked important recent literature on this topic includ- ing, for example, McNorton et al. (2022; doi:10.5194/acp-22-5961-2022) who used TROPOMI data to constrain methane emissions during COVID.*

An egregious oversight that we will address in the revised manuscript. We will also provide a narrative about the comparison of results.

*The authors don't seem to have reported uncertainties. It's clear what changes are actually substantial or within the noise. For example, the abstract lists changes of -3 Tg and -5 Tg as "substantial" in the abstract (Line 18). These don't seem particularly large. The text later claims that their work is within the uncertainty of another paper (Line 119), so it would be good to see uncertainties reported throughout.*

Good point. We have now included the uncertainties for our regional changes in posterior emission estimates.

**4.3 Error correlations**

*Where do the temporal and spatial prior error correlations come from? It seems that the authors use spatial correlation lengths of 300 km and 1 month. Are these important in the spatial patterns found here?*

These are based on our previously published studies for which we show that using these correlation length scales are not important to the large-scale emission distribution we report in this manuscript.

**4.4 Introduction**

*This reviewer is a bit confused by the list of citations in the intro. Specifically, Lines 34-35. The authors claim there is an intense debate on the role of fast growth in 2020 and 2021. They then claim that work has shown the importance of regional anomalies in the tropics. But many of these studies are from earlier than the time period being discussed.*

*"The underlying reasons for these anomalous growth rates in 2020 and 2021 are cur- rently subject to intense debate with some studies attributing most of the growth in 2020 to a reduction in the hydroxyl radical (OH) sink of methane due to global-scale reductions in nitrogen oxides due to pandemic-related industry shutdowns (Laughner et al. 2021). On the face of it, this appears to be a reasonable explanation, but recent stud- ies have used satellite observations of atmospheric methane to reveal regional hotspots over the tropics that are responding to changes in climate and have global significance (Pandey et al. 2021; Lunt et al. 2019; 2021; Pandey et al. 2017; Feng et al. 2022; Palmer et al. 2021; Wilson et al. 2020)."*

The reviewer is correct. In the revised manuscript, we will broaden the argument being outlined here. We focused on 2020 but we should have discussed the broader OH/emission argument, and the different data being used to make those points.

**4.5 Correlative data**

*The authors show plots of the changes in correlative data, but don't show spatial correla- tions. In this reviewer's opinion, it would be helpful to show a map with the correlation between the emission anomalies and the correlative data. The manuscript currently requires the reader to make the connection themself.*

We report correlations between different regions, as this reviewer has noted. We found that spatial distributions are more difficult to report based on two years of data, due to the coarse spatial resolution of our posterior emission estimates and due to regional time lags associated with rainfall and methane emissions. Consequently, we decided it would not be a useful addition to the manuscript. We will explain in the revised manuscript why we do not include this figure.

**References**

Feng, L., Palmer, P.I., Zhu, S. *et al.* Tropical methane emissions explain large fraction of recent changes in global atmospheric methane growth rate. *Nature Comm* **13,** 1378 (2022). https://doi.org/10.1038/s41467-022-28989-z

Lunt, M. F., Palmer, P. I., Feng, L., Taylor, C. M., Boesch, H., and Parker, R. J.: An increase in methane emissions from tropical Africa between 2010 and 2016 inferred from satellite data, Atmos. Chem. Phys., 19, 14721–14740, https://doi.org/10.5194/acp-19-14721-2019, 2019.

Lunt, M. F., P. I. Palmer et al, Recent rain-fed pulses of methane from East Africa during 2018-2019 contributed atmospheric growth rates, 16(2), Env. Res. Lett., https://doi.org/10.1088/1748-9326/abd8fa, 2021.

Sun, W., Zhu, L., De Smedt, I., Bai, B., Pu, D., Chen, Y., et al. (2021). Global significant changes in formaldehyde (HCHO) columns observed from space at the early stage of the COVID-19 pandemic. *Geophysical Research Letters*, 48, e2020GL091265. https://doi.org/10.1029/2020GL091265

Wilson, C., Chipperfield, M. P., Gloor, M., Parker, R. J., Boesch, H., McNorton, J., Gatti, L. V., Miller, J. B., Basso, L. S., and Monks, S. A.: Large and increasing methane emissions from eastern Amazonia derived from satellite data, 2010–2018, Atmos. Chem. Phys., 21, 10643–10669, https://doi.org/10.5194/acp-21-10643-2021, 2021.

---

## Author Comment (AC3)

We thank both reviewers for providing useful comments to the submitted manuscript. We also thank Janne Hakkarainen for spotting a typo in Appendix B that we have now fixed. Below we detail our responses to individual comments and, if relevant, how we altered the revised manuscript.

**Response to reviewer 1**

Our initial response was in late June 2022. Here we include more detailed responses

**Major comments**

*A crucial aspect of this work that is missing is the evaluation. There seems to be no evaluation of the results using independent observations or techniques like k-fold cross validation. The work addresses an important and timely topic but, in this reviewer's opinion, the main claims in the manuscript (that the growth is driven by emissions, not chemistry) do not seem supported by their numerical experiments.*

We have indirectly evaluated our results for 2020 by inferring methane emissions using the NOAA surface data, which provide consistent results on global and continental spatial scales. We will make this point clearer in the revised manuscript. When we first responded to this comment in June 2022, we did not have access to NOAA data for all sites in 2021.

We have now evaluated our posterior emissions using independent TCCON XCH4 measurements (v GGG2020) for 2020 (Figure C1) and the first half of 2021 (not shown). We find that the annual mean model minus TCCON XCH4 values for 2020 are within 10 ppb for 16 (out of 18) sites with standard deviation between 5 and 18 ppb but typically < 10ppb. This is consistent with our evaluation of data from 2009-2019 (Feng et al, 2022), building on previous studies that have included substantial evaluation using a range of in situ and remote sensing data. Our preliminary analysis of an incomplete year of TCCON data in 2021 shows similar model performance. To address this point, we have added Appendix C that includes a figure showing the mean comparison statistics.

The suggestion to use k-fold cross validation is interesting. We have used this method in other less computationally intensive applications but to our knowledge it has not been used to evaluate a global inversion. We are unconvinced this will add much to our narrative, especially given the focus of the review is on 2020.

*The major concern this reviewer has with the manuscript is that the title and central claims don't seem supported by their data. The main scientific claim (and their final conclusion) is that the record-breaking methane growth rates in 2020 and 2021 were driven by emissions, not chemistry. This claim certainly seems plausible (if not likely), but their experiments do not seem suficient to justify that claim. In this reviewer's opinion, the ideal way to conclude as to the relative importance of emissions and chemistry would be to include both emissions and OH in the state vector for their EnKF. That would result provide a straight forward assessment of the relative role of each process.*

The sensitivity experiment we used to quantify the impact of OH is admittedly a brute-force approach but in the absence of rigorous constraints on OH concentrations we felt this was a transparent approach that is easy to understand.  As part of our revision, we performed new experiments that assume that the OH reduction in 2020 roughly followed the observed temporal and spatial changes for tropospheric ozone (Ziemke et al., 2022). Again, we find

that increased emissions largely explain the large global mean growth of atmospheric methane in 2020. This is described in Appendix D.

An alternative way would indeed be to include both emissions of methane and OH in the state vector, and this is what we have done in the revised manuscript. These additional calculations are described in Appendices D and E and summarized in the main text.

We have conducted new experiments by including monthly scaling factors for OH concentrations over 18 global regions as part of state vector (Appendix E). The resulting emission increase between 2019 and 2020, is now about 25% smaller than our control run that solves for methane emission estimates using OH climatology. This result is consistent with emissions being primarily responsible for the anomalous global atmospheric methane growth rate in 2020. As the reviewer will be aware there are also still gross assumptions associated with this OH-methane emission inversion and the resulting posterior results do not provide a "straightforward assessment" of OH changes due to the limited information content of the satellite and in situ atmospheric methane data. Methane inversions that also estimate OH, typically quantify OH on large spatial scales and therefore will be unable to identify localized changes in OH that we anticipate happened during the Covid-related lockdowns.

The ideal approach would be to include the methane emissions in a state vector as part of a full-chemistry data assimilation system that also consider more appropriate constraints on OH. But of course, this problem becomes progressively more complex (non-linear) and more intractable (associated with more chemistry constituents and more data).

See also our response below, which discusses larger OH perturbations.

***The argument presented in this manuscript, as this reviewer interpreted it, is as follows:***

***the authors conducted a global inversion at 2◦ × 2.5◦ resolution with an EnKF from 2019-2022. This inversion assumes constant OH fields for the 3-year window. The authors find changes in the magnitude and spatial patterns of methane emissions. the authors compared these emission changes to rainfall, GRACE groundwater, and temperature. The largest correlations were 0.5-0.6 (representing 25–35% of the variability). the authors conducted a second global inversion with the same setup but reduced OH by 5% through 2020, where the largest COVID changes occurred.***

We are certainly more confident in the geographical distribution of emissions rather than their attribution. A more detailed study about the attribution of those emissions will be forthcoming but it is outside the scope of this current study.

Saying that, the spatial and seasonal variations in tropical methane emissions are consistent with those reported for earlier years (e.g., Lunt et al, 2019, 2021; Feng et al 2022; Wilson et al, 2021) that showed the response of methane emissions was consistent with microbial sources. However, we think imperfect attribution does not detract from our key message: large emissions are predominantly responsible for anomalous atmospheric growth rates in 2020 and 2021.

***The authors show the difference in emissions resulting from these cases but it is not clear to this reviewer which result is better. The differences seem to be central to their conclusions as indicated in the last two lines of their abstract ("Based on a sensitivity study for which we assume a conservative 5% decrease in hydroxyl concentrations in 2020...we find that the global increase in our a posteriori emissions in 2020 is ~22%***

*lower than our control calculation. We conclude therefore that most of the observed increase in atmospheric methane during 2020 and 2021 is due to increased emissions.") but I could not discern how they concluded why one was better than the other. Specifically, it is unclear why the control calculation is the correct answer here.*

We did not conclude one was better than the other. We concluded that a 5% drop in OH was too large using our perturbation approach, which described only 22% of the emission from our control run. Consequently, most of the atmospheric growth in 2020 and 2021 was due to emissions.

Our additional sensitivity calculations, including an OH inversion, are all consistent with our original hypothesis about increasing emissions in 2020 being the most likely culprit for the unprecedented global atmospheric methane growth rate. We will also put this discussion in the context of higher, localized reductions in OH as raised as a discussion point by this reviewer (see below).

*Evaluation and/or overfitting*

*Two common methods for evaluating the performance of optimization schemes are to: 1) evaluate against independent observations or 2) perform k-fold cross validation. Neither of these were included here. This is something that should be included for all their cases with an inversion analysis to ensure that one is not overfitting for a particular inversion*

This study is a two-year extension of recent work (Feng et al, 2022), including substantial evaluation of the results that build on previous studies. In an earlier response (see above) we have described our evaluation with TCCON data for 2020. We have also indirectly evaluated our 2020 results using an inversion constrained by NOAA data that leads to consistent results.

Since we use an ensemble Kalman filter the error characterization is a natural extension of our analysis. Using the cost function weights, we find no evidence to suggest we have overfitted the data. We have added text to that effect in the revised manuscript.

*OH is inconsistent with other work*

*The authors chose a 5% reduction in OH based on Laughner et al. (2021). However, Laughner et al. (2021) was a review/synthesis paper that took global mean OH changes from Miyazaki et al. (2021; doi:10.1126/sciadv.abf7460) and used them in a box model. This reviewer wonders how large the global mean OH changes are in this manuscript from Feng et al.? My suspicion is that they are quite a bit smaller than what was reported in Miyazaki et al. and Laughner et al Additionally, the OH chemistry is highly non-linear and Miyazaki et al. discuss how OH and ozone actually increase in some regions despite the NOx reductions. Using OH fields from Miyazaki et al. would be a much better way of testing if the OH simulated in that work impacted the methane burden. Essentially, this reviewer does not think the OH sensitivity run designed here accurately portrays the OH changes that others have found. Data supporting the choice of OH runs used here would help assuage these concerns.*

Our sensitivity experiments are designed to examine how large-scale reductions in OH (associated with Covid-19 lockdowns) affects our methane emission estimates inferred from atmospheric methane measurements.

First, it was remiss of us not to include Miyazaki et al, 2021 for which we apologize. This primary reference was added to the revised manuscript. Figure S8A from this paper shows

localized OH reductions of 15-20% for May 2020, although the authors note that this reduction can be as large as 30% on a grid basis. This represents one of the early months of the shutdown when we expect the largest reduction in emissions, so we expect global mean annual changes to be smaller. We also note that Miyazaki et al, 2021 (as with others) do not consider the coincident changes in non-methane hydrocarbons (NMHCs) that will also affect non-linear ozone chemistry, including the production and loss of OH. Large coincident reductions in formaldehyde, for instance, a proxy of NMHCs, have since been reported in other studies (e.g., Sun et al, 2021).

We did a number of simulations in preparation for this manuscript. We used 1% and 5% global reductions in OH but penalized regions that would not have been impacted by reductions in nitrogen oxide emissions from Covid. We instead chose to reduce OH only over regions where there are substantial anthropogenic $CO_2$ emissions, resulting in a distribution similar to Miyazaki et al, 2021.

In response to this reviewer comment, we prepared a calculation in which we decrease OH by an additional 25% of eastern China for March 2020 (the peak of emission reductions). We find that this month-long perturbation can explain an additional 0.26 Tg of methane emissions for 2020, providing some indication of how a larger, localized reduction in OH will impact global changes in methane emission estimates. In the revised manuscript, we have reported the result of an additional calculation (Appendix D) that adopts an OH reduction pattern following changes in tropospheric ozone (Ziemke et al, 2022). We find the result is similar to the results from our other sensitivity test – the emissions needed to explain the atmospheric growth of methane in 2020 is about 25% lower than our control when we consider a drop in OH.

We have added a broader discussion of this kind of OH perturbation (Appendix D) and now included an OH inversion (Appendix E) in the revised manuscript. However, we hasten to add that this reviewer comment should be tempered by the uncertainties associated with previous studies that do not accurately describe the photochemical perturbation associated with the Covid-19 shutdowns. We added text that describes caveats associated with both approaches.

***GOSAT proxy observations***

***The authors use GOSAT proxy observations. This means that the methane concentrations will be dependent on the CO2 concentrations. However, it seems like the authors use CO2 simulations with monthly emissions through 2019. Therefore the CO2 could lead to a bias in their methane concentrations during COVID due to the reduction in CO2 emissions. This would be most pronounced in urban areas.***

As described in Parker et al., 2020, the GOSAT data use an ensemble of $CO_2$ models based on atmospheric in situ inversions. For recent years, when updated model inversions based on in situ observations are not typically readily available on the timescales required for data processing, we have used $CO_2$ values from previous years that have been incremented by the NOAA global growth rate. To ensure that this does not introduce an error due to model $CO_2$ values, an inversion is performed that directly uses the XCH4/XCO2 ratio, without relying (explicitly or implicitly) on the model CO2. We have clarified this point in the revised manuscript.

***3.7. This reviewer was also very confused by the description of the data used in places. For example, when describing a sensitivity study the authors mention using proxy GOSAT XCH4 data (Line 90) but the main inversions also seem to use proxy GOSAT data.***

We use two approaches: 1) the proxy ratio (XCH4/XCO2) directly without need to assume a prior model for $XCO_2$ to multiply out to obtain XCH4, which is described in Feng et al, 2022; and 2) the proxy methane XCH4. In addition, we have also assimilated in situ methane data in both approaches. As a result, we find both approaches lead to generally consistent results. We will amend the text to make this point clearer.

**Minor Comments:**

**Oversight of previous work**

***The authors seem to have overlooked important recent literature on this topic including, for example, McNorton et al. (2022; doi:10.5194/acp-22-5961-2022) who used TROPOMI data to constrain methane emissions during COVID.***

An egregious oversight that we have now addressed in the revised manuscript.

***The authors don't seem to have reported uncertainties. It's clear what changes are actually substantial or within the noise. For example, the abstract lists changes of -3 Tg and -5 Tg as "substantial" in the abstract (Line 18). These don't seem particularly large. The text later claims that their work is within the uncertainty of another paper (Line 119), so it would be good to see uncertainties reported throughout.***

Good point. We have now included the uncertainties for our regional changes in posterior emission estimates.

**Error correlations**

***Where do the temporal and spatial prior error correlations come from? It seems that the authors use spatial correlation lengths of 300 km and 1 month. Are these important in the spatial patterns found here?***

These are based on our previously published studies for which we show that using these correlation length scales are not important to the large-scale emission distribution we report in this manuscript.

**Introduction**

***This reviewer is a bit confused by the list of citations in the intro. Specifically, Lines 34-35. The authors claim there is an intense debate on the role of fast growth in 2020 and 2021. They then claim that work has shown the importance of regional anomalies in the tropics. But many of these studies are from earlier than the time period being discussed.***

The underlying reasons for these anomalous growth rates in 2020 and 2021 are currently subject to intense debate with some studies attributing most of the growth in 2020 to a reduction in the hydroxyl radical (OH) sink of methane due to global-scale reductions in nitrogen oxides due to pandemic-related industry shutdowns (Laughner et al. 2021). On the face of it, this appears to be a reasonable explanation, but recent studies have used satellite observations of atmospheric methane to reveal regional hotspots over the tropics that are responding to changes in climate and have global significance (Pandey et al. 2021; Lunt et al. 2019; 2021; Pandey et al. 2017; Feng et al. 2022; Palmer et al. 2021; Wilson et al. 2020).

The reviewer is correct. In the revised manuscript, we clarify the argument being outlined here. We focused on 2020 but we should have discussed the broader OH/emission argument, and the different data being used to make those points.

***Correlative data***

***The authors show plots of the changes in correlative data, but don't show spatial correla- tions. In this reviewer's opinion, it would be helpful to show a map with the correlation between the emission anomalies and the correlative data. The manuscript currently requires the reader to make the connection themself.***

We report correlations between different regions, as this reviewer has noted. We found that spatial distributions are more difficult to report based on two years of data, due to the coarse spatial resolution of our posterior emission estimates and due to regional time lags associated with rainfall and methane emissions. Consequently, we decided it would not be a useful addition to the manuscript. We explain in the revised manuscript why we do not include this figure.

**Response to review comment 2**

*General:*

***Feng et al attempts to attribute the stunning increases in the methane growth rate from 2020-2021 using satellite observations of methane in conjunction with surface data. The subject is very timely and worth getting out into the community for further analysis and discussion. However, there are a number of issues that the authors need to address, which are incorporated into the annotated text of the paper. In particular, there are no uncertainty estimates of the fluxes, no real explanation of the Eastern African increase, and an unrealistic description of OH change. Spending more time on these elements will strengthen the paper and adding credibility to the results.***

OH reduction is still challenging to quantify due to lack of (direct) observations. In the revised manuscript (Appendix D), we include additional inversion experiments based on different assumptions on 2020 OH reduction. In Appendix E, we estimate OH scaling factors (with the surface emissions of methane) from methane concentration observations. Our results all suggest that increased emissions in 2020 are primarily responsible for the large increase in the global atmospheric growth rate of methane.

Recent regional inversions based on both GOSAT and TROPOMI data also show large emission from Africa in 2020/2021 (not shown), and they are consistent with changes in hydrological data and land surface data. Further research is needed to understand the physical mechanisms responsible for the Eastern African emission changes, which is the outside the scope of this manuscript.

*By how much?*

By using a higher bias correction, we reduced the model overestimate compared to HIPPO aircraft data by about 5ppb in earlier years (2010-2011) prior to our experiment period.

**For both CO2 and CH4, or just CH4?**

It is just for $CO_2$ prior emission. For anthropogenic $CH_4$ emission, we have used EDGAR v4.32. We have restructured the paragraph so that the reader can easily find details about the CO2 and methane prior inventories.

**Olsen and Randerson is a downscaling technique. What biospheric model did you use?**

We used the results from the NASA CASA model (Randerson et al., 1996) which we now clarify.

**A 5% reduction is not that conservative. The spatial pattern of OH reduction is quite important. Miyazaki et al, 2021 (Science Advances) shows that the global mean reduction in OH is about 4%. However, the reductions can be substantially higher, (>30%) locally. These will directly affect the methane inferences, especially over Europe and parts of Asia. Please review the paper and perform a sensitivity analysis closer to reported spatial changes in OH.**

In our revised manuscript we performed two additional inversions with different (assumed) temporal and spatial patterns for the reduced OH, including

1) a higher (25%) OH reduction over China in March 2020.
2) global reduction pattern following observed ozone changes

*We could expect, however, that the CO2 emission in 2020 and 2021 to be different than 2019. Not sure how this really addresses the OH issue. As noted next, the Miyazaki et al, 2021 provides a more observationally constrained pattern of OH.*

Here we use $CO_2$ emission as proxy spatial pattern for the reduction of OH due to reduced human activity by COVID lockdown. In the revised manuscript we tested different assumptions for the OH reduction pattern and conducted a new inversion experiment by including OH scaling factors to the methane emission state vector to inferred from methane observations.

To echo our response to a similar comment from Reviewer 1, there are no effective constraints to provide accurate estimates for the spatial and temporal change in OH during the Covid lockdown.

**Is this a correlation in flux or concentration space? If it is the former, where did 300km come from?**

It is for surface emissions. We take this value from a previous study (Feng et al., 2022). It has no significant impact on the estimated global change of methane emissions between 2019 and 2020.

**You should be able to calculate the simulated atmospheric CH4 growth rate for 2020 and 2021 from your inversion. You should add that to the table (1).**

Good point. We estimate the global growth rate in 2019 is 6.7 ppb/yr, and 15.6 ppb/yr in 2020 from the simulation forced by posterior methane emissions inferred from GOSAT. There will be differences between the global growth rate inferred from GOSAT data and *in situ* data due to differences in geographical coverage.

*The box-model approach carries its own assumptions, and can't be really used to validate the CH4 topdown estimates.*

We use this model to show that based on a mass balance argument, satellite column XCH4 data show a larger net emission change between 2019 and 2020 compared to values inferred from the coarser in situ surface data.

**Based upon NOx emission changes during 2020-2021, we would not expect CH4 lifetime to be fixed.**

Please see our response to the previous question. Here we only the model to quantify the net emission change from mass balance. We agree that the methane lifetime may very well be different in 2020, and the implications of that our conclusions have been discussed in Appendix D and E.

*There are no uncertainty estimates in this calculation. I can not assess whether these changes are tatistically significant or not. Uncertainty estimates need to be added to these flux estimates. These could be done through comparison with other inversions or independent observations.*

We have added the uncertainties to regional fluxes.

**It looks like the increases in SA in 2021 are an acceleration of the 2020 increases. What is going on there?**

It is an interesting observation. Currently, we do not have sufficient independent data to explain this change.

**How is anomalous defined here? The temperature variations need to be calculated relative to the long term temperature variability to assess whether they are not within climatology.**

Those anomalies have been calculated based on the mean between 2010-2021 when GRACE data are available.

**Please put plots of those regressions in the supplemental.**

Good suggestion. We add the plot Figure A6 for correlation between monthly flux anomaly and GRACE LWE anomaly during 2018-2021.

**Cooper didn't discuss NOx emissions, only concentrations. The impact on OH would need to be calculated separately and would be affected by other reactive species, e.g., ozone.**

The reviewer is right. We have clarified this point and the impact on OH being more complicated.

**It's hard to see the differences in a spatial plot. Please make a difference plot using the regions used in Fig. For 2020 and 2021 between the proxy retrieval and the simultaneous estimate.**

We have added a new plot (Figure A7) in the revised manuscript, following this suggestion

*The implication of East Africa as the single largest driver of the methane growth rate is puzzling. Yes, there is a correlation with water anomalies, but it's not the largest driver of methane in driver. A more plausible set of physical mechanisms needs to be proposed for why we could expect this region as a dominant driver.*

This result is consistent with (and builds on) a number of preceding studies that have shown the relationship with rainfall, river flow down the Nile, and changes in the extent of the Sudd wetland. We have added Figure A6 that shows the strong relationship between liquid water equivalent anomalies inferred from GRACE and methane flux anomalies over the region. High resolution inversions enabled by TROPOMI have also highlighted that South Sudan is a globally significant source of atmospheric methane.

*This is not clear to me at all that this is a valid assumption. The OH responses will depend on the regional NOx emission reductions and the timing of those reductions, in part because it will also affect ozone production. There are better sources for OH than this crude assumption.*

We have now included an additional experiment that uses a different assumption to define the OH reduction pattern and have also included an OH inversion that infers scaling factors and methane emission estimates from atmospheric methane data. Our results all confirm the dominant role of increased emissions in explaining observed global growth of atmospheric methane.

**References**

1. Feng, L., Palmer, P.I., Zhu, S. et al. Tropical methane emissions explain large fraction of recent changes in global atmospheric methane growth rate. Nature Comm 13, 1378 (2022). https://doi.org/10.1038/s41467-022-28989-z 5
2. Lunt, M. F., Palmer, P. I., Feng, L., Taylor, C. M., Boesch, H., and Parker, R. J.: An increase in methane emissions from tropical Africa between 2010 and 2016 inferred from satellite data, Atmos. Chem. Phys., 19, 14721–14740, https://doi.org/10.5194/acp-19-14721-2019, 2019.
3. Lunt, M. F., P. I. Palmer et al, Recent rain-fed pulses of methane from East Africa during 2018-2019 contributed atmospheric growth rates, 16(2), Env. Res. Lett., https://doi.org/10.1088/1748-9326/abd8fa, 2021.
4. Sun, W., Zhu, L., De Smedt, I., Bai, B., Pu, D., Chen, Y., et al. (2021). Global significant changes in formaldehyde (HCHO) columns observed from space at the early stage of the COVID-19 pandemic. Geophysical Research Letters, 48, e2020GL091265. https://doi.org/10.1029/2020GL091265
5. Wilson, C., Chipperfield, M. P., Gloor, M., Parker, R. J., Boesch, H., McNorton, J., Gatti, L. V., Miller, J. B., Basso, L. S., and Monks, S. A.: Large and increasing methane emissions from eastern Amazonia derived from satellite data, 2010–2018, Atmos. Chem. Phys., 21, 10643–10669, https://doi.org/10.5194/acp-21-10643-2021, 2021.
6. Ziemke, J. R., Kramarova, N. A., Frith, S. M., Huang, L.-K., Haffner, D. P., Wargan, K., et al. (2022). NASA satellite measurements show global-scale reductions in free tropospheric ozone in 2020 and again in 2021 during COVID-19. *Geophysical Research Letters*, 49, e2022GL098712. https://doi.org/10.1029/2022GL098712

---

## Author Response (AR2)

**Author Responses to Second Round of Reviewer Comments**

We thank both reviewers for their additional comments. We have addressed all the comments received, including the clarification of several points and changing the manuscript title. We have now included a joint OH-methane inversion as suggested by Reviewer 1. On reflection we decided to report methane emission estimates inferred using OH climatology and the new inversion. This helps illustrates the value of including OH in the state vector on a routine basis. We have also put our results into context of work that has been published since we submitted our original manuscript.

*Reviewer 1*

*First, I think the manuscript is greatly improved and I commend the authors for both adding additional inversions and using simulations that more accurately represent previous work. However, in this reviewer's opinion, the paper needs to be restructured. I am generally reluctant to suggest structural changes to a manuscript, but the current structure resulted in both confusion and misunderstandings by this reviewer about what the authors \*actually\* did and what they report. Frankly, this reviewer is still confused about which numbers are being reported in the abstract and what contributes to their uncertainties. This reviewer feels the manuscript still needs major revisions before being publishable.*

We thank the reviewer for these additional comments. We have absorbed the text from Appendices D-F into the main text. We have retained Appendices A (supplementary figures), B (box-model calculation), and C (evaluation) to promote readability of the study unencumbered by excessive figures. In doing so, we believe the paper now presents the OH sensitivity calculations more clearly, summarized in Table 2.

*To start, this manuscript would greatly benefit from a Table that explains the various simulations including what was held constant, what was perturbed, etc. It could list both 3D inversions, forward simulations, and the Box Model. This should follow a general description of the methodology.*

Agreed. We have included Table 2 that summarized the control and OH sensitivity calculations.

*The material in the FIVE appendix sections should be moved into the main text. Much of this material is central to their conclusions and, in this reviewer's opinion, should not be relegated to the appendix. The main text references an OH sensitivity run, but the manuscript now includes at least 3 different descriptions of OH: their 5% OH change over CO2 source regions, OH described in Appendix D, and OH inferred in Appendix E.*

Agreed, we have now included the experiment descriptions and results in the main paper.

*Following this, it is not clear to this reviewer why the authors still include the 5% OH change over CO2 source regions. Both reviewers criticized this sensitivity run because it does not represent what previous work found. Yet this is currently the only OH sensitivity run that is described and discussed in detail in the main text.*

We agree that this approach represents an idealised situation but do provide useful insights especially since we have no definitive way of quantifying OH changes during 2020. We have de-emphasized these calculations, but they now provide a useful sanity check for interpreting results from the joint OH-methane inversion, and we now describe in that way. We find our idealized calculations are broadly consistent on the global scale with the joint OH-methane inversion.

*Regarding the reported numbers and conclusions, it seems that the authors should be reporting results from their joint inversion of both OH and methane emissions. This seems like the numerical experiment that allows them to make a quantitative statement about the relative importance of sources and sinks, which is the central claim of the manuscript. However, from my reading of the manuscript, I do not think this is the numerical experiment that is used for the numbers in their abstract/conclusions, although I am not actually sure which experiment their numbers come from.*

We think we have now struck a balance. We have included the *ad hoc* sensitivity calculations with the OH inversion to build a narrative. We feel that presenting a new methane-OH inversion as a *fait accompli* would not be instructive in this case, and we believe it is much stronger as presented in the context of the fixed-OH baseline calculations and the other sensitivity experiments that produce consistent results. We have, as described above, de-emphasized the ad hoc sensitivity calculations and focused on the results inferred from the formal joint OH-methane inversion.

*Regarding the box modeling, does the box modeling reflect the updated findings regarding OH? It seems that the text was not updated even though the authors state: "after considering the effects of methane sinks, we find that a one-box model calculation…" (Line 133).*

*It is still not clear how much the authors are perturbing global mean OH in their sensitivity runs. This would be important to state. It seems that it is still less than other work reports.*

The purpose of the box model calculation was exclusively to help explain the different atmospheric methane growth rates inferred from GOSAT and *in situ* data. We show that using different observation coverage, the resulting estimates for the atmospheric methane growth agree with each other on a multiple-year timescale, but not necessarily individual years. It does not include the influence of OH. The box model calculation explains why by using *in situ* data Peng et al have overestimated the influence of OH in 2020 even though our studies agree on the change in the OH. Similarly a recent study (Qu et al., 2022) based on GOSAT XCH4 retrievals also show larger atmospheric methane increase between 2019 and 2020 than reported by Peng et al. (2022). We have updated the concluding remarks to reflect that point.

*I have a number of questions regarding the OH inversion as I think this is the numerical experiment that actually supports their conclusions and claims:*

*- What do the spatial patterns of the OH inversion look like?*

*- What about the temporal pattern?*

First, we have reduced the state vector so we only report annual OH changes on six 25-degree latitude zonal band. We find that the data can support the independent inference of methane emissions and OH changes over those broad regions. We have added Figure 5 to make that point, which shows that *a posteriori* correlations between OH and methane emission regions are typically < 0.1. Figure 7 shows the annual difference in *a posteriori* methane loss due to OH variations in 2020 and 2021 compared to the inversion that uses OH climatology. We describe those changes in Section 3.

*- The authors mention that results in Appendix D and E are consistent but provide no numbers, figures, or really anything to back up that claim. How was this assessed? What is the bar for "consistency"?*

Agreed that is confusing. Consistency in this example means that our sensitivity experiments all point to OH representation less than 30% of the observed atmospheric growth rate of

methane. This statement remains valid and we have clarified this point, although the sensitivity studies has been de-emphasized in the manuscript.

*- The basis functions for OH differ from methane, does that matter?*

It is an interesting question, but the data does not contain sufficient information to match the resolution of the methane emission state vector. In the revised manuscript, we have simplified the OH state vector to six 25-degree zonal bands, which we find can be supported by the data (Figure 5). As part of the preparatory calculations for the manuscript, we explored using more scaling factors and we didn't find a significant difference to the result we presented in the OH change between 2020/2021 and 2019 but revealed stronger correlations between state vector elements.

*- The authors mention that we do not have sufficient constraints for OH, yet they aim to solve for both longitudinal and latitudinal changes. Are those well constrained? Why not just solve for OH as a function of latitudinal bands?*

Yes, this is a great point. On reflection we further simplified our OH state vector and describe in the manuscript the performance of the joint OH-methane inversion. Figure A5 shows we can, indeed, solve independently regional methane emission and zonal band changes in OH.

*Another paper was recently published in Nature that uses similar data and reaches similar conclusions (~50/50 sources and sinks; https://www.nature.com/articles/s41586-022-05447-w). This work should be mentioned as it is directly relevant. This paper also included both methane and OH in the state vector (as did Zhen et al., 2022). Therefore I stand by my earlier review that the bar for claiming emissions are responsible for the changes necessitates inverting for both methane and OH*

We have now cited this paper in the concluding remarks. Peng et al use *in situ* data so as we now discuss in our paper this introduces a negative bias in emission increase estimates between 2020 and 2021, and consequently a positive bias in the influence of OH on the atmospheric growth rate in 2020; our results for reduced OH in 2020 are remarkably consistent with those reported by Peng et al.

To clarify, Peng et al, do not include OH and methane together in their state vector to determine their top-down methane flux estimate. First, they calculate the magnitude and changes in OH distributions due to reduced emissions using the LMDZ-INCA model and infer OH changes by fitting a 12-box model to HCFC-141b, HFC-32, and HFC-134a measurements. This calculation will of course be subject to errors in inventories but it is reassuring that we get similar results for the reduction in OH.

We are not familiar with Zhen et al, 2022. But we can say is that our paper in its current form does provide different lines of evidence, including a joint methane-OH inversion, that are consistent with increased emissions playing the major role in the atmospheric growth rate in 2020.

---

## Author Response (AR3)

**Author Responses Technical Corrections**

The errors were associated with cross referencing to Figures and Tables that were collected in a separate file. We have now fixed this issue.

PIP 29th March 2023.